# Probabilistic computing with NbO$_x$ metal-insulator transition-based self-oscillatory pbit

Hakseung Rhee [1], Gwangmin Kim [1], Hanchan Song [1], Woojoon Park[1], Do Hoon Kim[1], Jae Hyun In [1], Younghyun Lee [1] & Kyung Min Kim [1] ✉

Energy-based computing is a promising approach for addressing the rising demand for solving NP-hard problems across diverse domains, including logistics, artificial intelligence, cryptography, and optimization. Probabilistic computing utilizing pbits, which can be manufactured using the semiconductor process and seamlessly integrated with conventional processing units, stands out as an efficient candidate to meet these demands. Here, we propose a novel pbit unit using an NbO$_x$ volatile memristor-based oscillator capable of generating probabilistic bits in a self-clocking manner. The noise-induced metal-insulator transition causes the probabilistic behavior, which can be effectively modeled using a multi-noise-induced stochastic process around the metal-insulator transition temperature. We demonstrate a memristive Boltzmann machine based on our proposed pbit and validate its feasibility by solving NP-hard problems. Furthermore, we propose a streamlined operation methodology that considers the autocorrelation of individual bits, enabling energy-efficient and high-performance probabilistic computing.

The hyper-connected era, characterized by the Internet of Things, big data, and artificial intelligence, demands efficient computing solutions for solving combinatorial optimization problems such as route search, network optimization, etc[1]. However, these problems are often non-deterministic polynomial-time-hard (NP-hard), posing a challenge for conventional deterministic computing, which requires vast resources and yields incorrect local minimum solutions[2,3]. Energy-based computing has emerged as a potential solution to this challenge. One example is adiabatic quantum computing (AQC), which encodes problems into energy landscapes and leverages quantum mechanics to identify the lowest energy state corresponding to the correct answer[4,5]. AQC can effectively achieve global minima by escaping local minima with quantum-mechanical principles. However, its requirement for an ultra-low temperature environment limits its application to edge devices.

Probabilistic computing (p-computing) has recently emerged as a promising energy-based computing system. Unlike other approaches, p-computing is operable at room temperature and compatible with CMOS technology[6–9], making p-computing highly feasible and attainable. It employs probabilistic bits (pbits), which fluctuate probabilistically between 0 and 1, like the probabilistic behavior of qubits. The first CMOS-compatible pbit device was proposed using a magnetic tunnel junction (MTJ) structure[10]. The MTJ cell possesses two energetically equal states (i.e., parallel and antiparallel spins) separated by an energy barrier. As this energy barrier is sufficiently low so that the intrinsic thermal noise flips its state, it results in fluctuation between the two states. In addition, the energy level of each state can be controlled by externally applied voltage, thereby modulating the probability of having a particular state. This work implies that any physical systems exhibiting bi-stability can potentially be used as pbits for energy-efficient computing.

Metal-insulator transition (MIT) in transition metal oxides such as NbO$_x$ or VO$_x$ is a phenomenon that exhibits bi-stability between the metal and insulator phases at a certain transition temperature ($T_{MIT}$)[11].

[1]Department of Materials Science and Engineering, Korea Advanced Institute of Science and Technology (KAIST), Daejeon 34141, Republic of Korea.
✉e-mail: km.kim@kaist.ac.kr

MIT dynamics are highly complex, involving electrical and thermal dynamics coupling, which can be used in various emerging physical computing devices, such as biomimetic/neuromorphic artificial intelligence and cryptography devices[12–14]. The MIT at the $T_{MIT}$ can be easily disturbed by slight irregularities, making the device offer pbit functionality. In addition, the $T_{MIT}$ can be attained by Joule heating[15–18], allowing for electrically modulable thermal dynamics. Furthermore, with a series resistor, NbO$_x$ volatile memristors generate oscillating current outputs under a direct current (DC) bias[19,20].

Thus, by combining the probabilistic behavior of bi-stability with the oscillation characteristics, it is possible to obtain the probabilistic oscillation, which can be potentially used as a new type of pbits. Moreover, differing from conventional pbits[9,21,22], such oscillator-based pbits can generate a self-sustaining bitstream without bit-generating signal pulses. This can reduce power consumption and increase the stability of the system. Therefore, it is worthwhile to investigate developing the pbit from the metal-insulator transition and evaluate its potential for next-generation p-computing.

In this study, we propose an oscillatory pbit device embodying an NbO$_x$ volatile memristor. We observed that the oscillation is probabilistic when the MIT is involved during oscillation. Furthermore, the oscillation probability ($p_{osc}$) is controllable by modulating the $V_{ext}$, resulting in a sigmoidal $p_{osc}$-$V_{ext}$ relation suitable for p-computing. We also propose a model that accurately reproduces the experimental results, indicating that thermal and electrical noises trigger the probabilistic oscillation. Then, we demonstrate a memristive Boltzmann machine to validate its p-computing capability by solving graph-based combinatorial optimization problems. Lastly, we present the inherent autocorrelation issue and propose solutions for energy-efficient problem-solving.

## Results

### Probabilistic oscillation of NbO$_x$ oscillator

Figure 1a shows the current-voltage ($I$-$V$) behavior (black line) of the NbO$_x$ volatile memristor (TiN/NbO$_x$/TiN-via) measured from a current sweep from 0 to 1 mA. For the device integration, a square-shaped 40 nm-width TiN-via bottom electrode had been prepared from a commercial foundry. Then, a 20 nm-thick NbO$_x$ layer and a 50 nm-thick TiN top electrode were deposited by reactive sputtering (device structure is shown in the inset). The device capacitance was 6.47 pF and it exhibited repeatable double negative differential resistance (NDR) behaviors. The constant device capacitance and almost identical $I$-$V$ characteristics after electroforming with respect to the device's area (Supplementary Fig. S1) suggested that the switching is associated with a localized region, following a core-shell model[15–18,20,23,24]. The first NDR (NDR-1) at a low current (from 0.03 to 0.07 mA) is accompanied by the thermally activated conduction mechanism, and box-shaped second NDR (NDR-2) at a high current range (from 0.55 to 0.8 mA) is attributed to the MIT of NbO$_x$. During the DC $I$-$V$ sweep, the operating current was sufficiently low not to involve non-volatile resistance changes caused by the electrochemical reaction, such as oxygen vacancy formation[25,26]. As shown in Fig. 1b, when a resistor ($R_L$) is connected serially, the circuit comprises a so-called self-oscillator and generates current oscillations under the DC voltage input ($V_{ext}$). Interestingly, the oscillating output current exhibited probabilistic behavior at a certain $V_{ext}$. The colored dashed lines in Fig. 1a indicate the load lines with $R_L = 1.7$ k$\Omega$ and $V_{ext} = 1.32$ V (blue), 1.38 V (red), and 1.44 V (violet), whose intersections are operating points giving different types of oscillating outputs.

Figure 1c plots oscillating probabilities ($p_{osc}$) as a function of the $V_{ext}$ from 1.32 to 1.46 V with a 0.01 V interval. The $p_{osc}$ is calculated by dividing the number of observed oscillation peaks ($N_{obs}$) by the

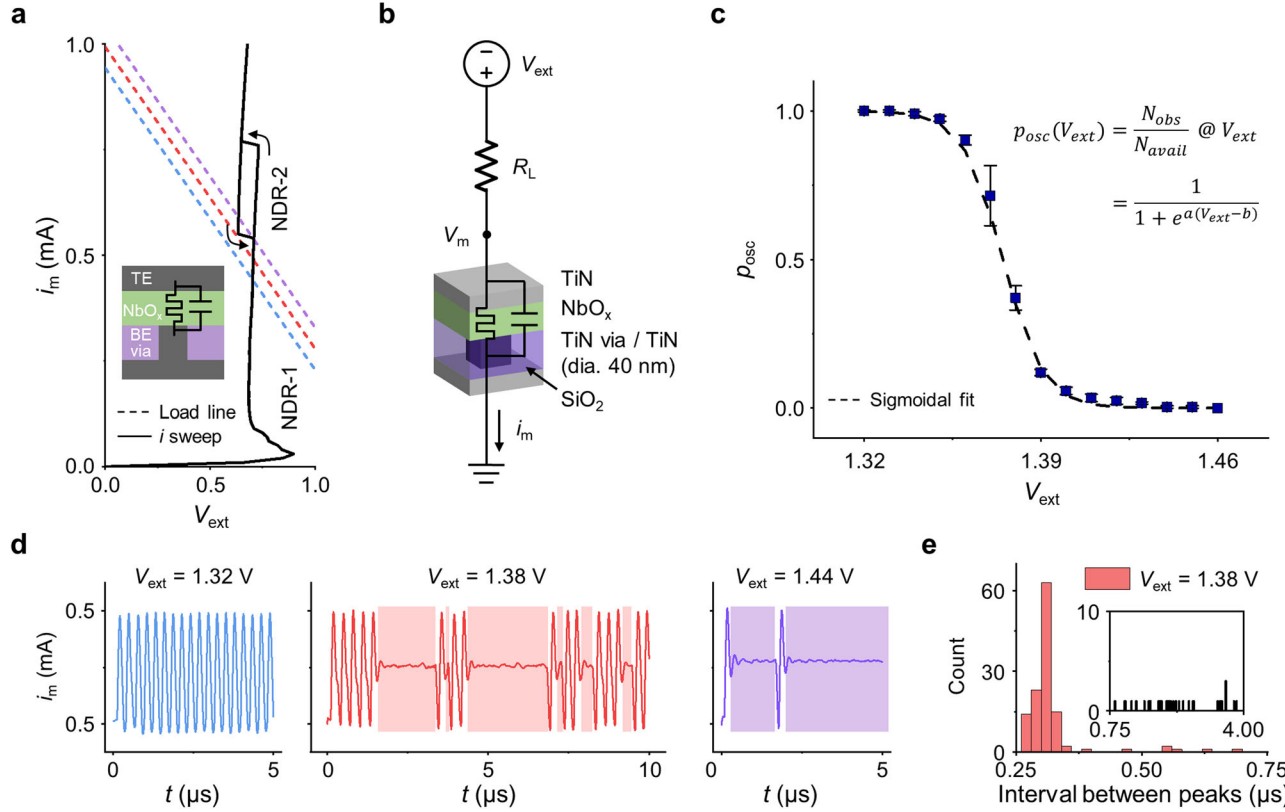

**Fig. 1 | Probabilistic oscillation behavior of a NbO$_x$ volatile memristor-based self-oscillator. a** An $I$-$V$ curve of the NbO$_x$ memristor (TiN/NbO$_x$/TiN-via) with a current sweep mode and load lines of $R_L = 1.7$ k$\Omega$. **b** Schematic of NbO$_x$ oscillator circuit and layer structure. **c** Oscillation probability ($p_{osc}$) distribution to the external voltage ($V_{ext}$) from 1.32 to 1.46 V with a 0.01 V interval. Error bars show standard deviation. **d** Three representative oscillation samples with varying $V_{ext}$; $V_{ext} = 1.32$ V ($p_{osc} = 1$), 1.38 V ($p_{osc} = 0.71$), and 1.44 V ($p_{osc} = 0.003$). **e** Histogram of the peak interval from 5 datasets in case of the $V_{ext} = 1.38$ V.

number of available peaks at the given frequency ($N_{avail}$). Each $p_{osc}$ value was an average from 5 datasets recorded for 30 μs (Supplementary Fig. S2). The probabilistic oscillation generates probabilistic bits in 260 ns with an energy of 114 pJ per oscillation or 141 pJ per non-oscillation (staying in the metallic state), which is faster than reported diffusive memristor-based[9]. Furthermore, the self-clocking nature allowed a compact implementation of the pbit in the circuit[12]. The $V_{ext}$-$p_{osc}$ curve fits well with the sigmoidal function, $p_{osc} = 1/(1 + e^{a(V_{ext} + b)})$ showed that the probabilistic oscillation satisfied the requirements of the pbit; the occurrence of oscillation at a certain time could be probabilistically 0 or 1, where the probability could be controlled by $V_{ext}$. Figure 1d shows three distinct oscillation behaviors with varying $V_{ext}$ as Fig. 1a; $V_{ext} = 1.32$ V ($p_{osc} = 1$), 1.38 V ($p_{osc} = 0.71$), and 1.44 V ($p_{osc} = 0.003$). At $V_{ext} = 1.32$ V, the NbO$_x$ oscillator generates periodic oscillation. Whereas, at $V_{ext} = 1.38$ V and 1.44 V, it generates irregular, probabilistic oscillation, with varying intervals (colored boxes) between peaks. Figure 1e plots the peak interval distribution of the $V_{ext} = 1.38$ V. The interval ranges from 260 ns to 3.8 μs, indicating that the probabilistic oscillation is irregular and random. More details on the measurement system configuration and its influence on the device characterization can be found in Supplementary Information section 3.

## Compact modeling of NbO$_x$ probabilistic oscillator

Ideally, the self-oscillator should either generate a periodic oscillation or not. The probabilistic oscillation suggests that some irregular dynamics are present, perturbing the deterministic behavior[27–30]. To identify the factors leading to this new type of non-ideal phenomenon, we designed a probabilistic oscillator (p-osc) model based on the deterministic model suggested by Suhas Kumar et al.[13,14,31] with the Ornstein-Uhlenbeck (OU) process[32].

The adopted deterministic model comprises a three-dimensional Poole-Frenkel conduction model (Eq. 1), Newton's cooling law (Eq. 2), a nonlinear function of thermal resistance describing the metal-insulator transition (Eq. 3), and Kirchhoff's law (Eq. 4) in the circuit of Fig. 1b. These equations are described as follows:

$$i(T,v) = \left[ \sigma_0 e^{-\frac{E_a}{2k_b T}} A \left\{ \left( \frac{k_b T}{\omega} \right)^2 \left( 1 + \left( \frac{\omega\sqrt{v/d}}{k_b T} - 1 \right) e^{\frac{\omega\sqrt{v/d}}{k_b T}} \right) + \frac{1}{2d} \right\} \right] v \quad (1)$$

$$\frac{dT}{dt} = \frac{1}{R_{th}(T)C_{th}} \left\{ R_{th}(T)i(T,v)v - (T - T_{amb}) \right\} \quad (2)$$

$$R_{th}(T) = \frac{R_{th,I}}{1 + \exp(-\beta(T_{MIT} - T))} + \frac{R_{th,M}}{1 + \exp(-\beta(T - T_{MIT}))} \quad (3)$$

$$\frac{dv}{dt} = \frac{1}{R_L C_m} \{ V_{ext} - R_L i(T,v) - v \} \quad (4)$$

where $A$ is the lateral area of the cell, $d$ is the thickness, $k_b$ is the Boltzmann constant, and $\omega$ and $\sigma_0$ are material constants. $T_{amb}$ is an ambient temperature, $C_{th}$ is a thermal capacitance, $R_{th}$ is a temperature-dependent thermal resistance composed of an insulating phase ($R_{th,I}$) and a metallic phase ($R_{th,M}$), and $T_{MIT}$ is a metal-insulator transition temperature. $C_m$ is an electrical capacitance of the NbO$_x$ layer (Table S2). Figure 2a shows the $I$-$V$ curve (black) and load line (blue dotted) obtained from Eqs. 1–4, where $dT/dt = 0$, $dv/dt = 0$, $R_L = 1.2$ kΩ, and $V_{ext} = 1.41$ V. The temperature sweep curve (red line) crosses the box-shaped $I$-$V$ curve of the NDR-2 region, meaning the

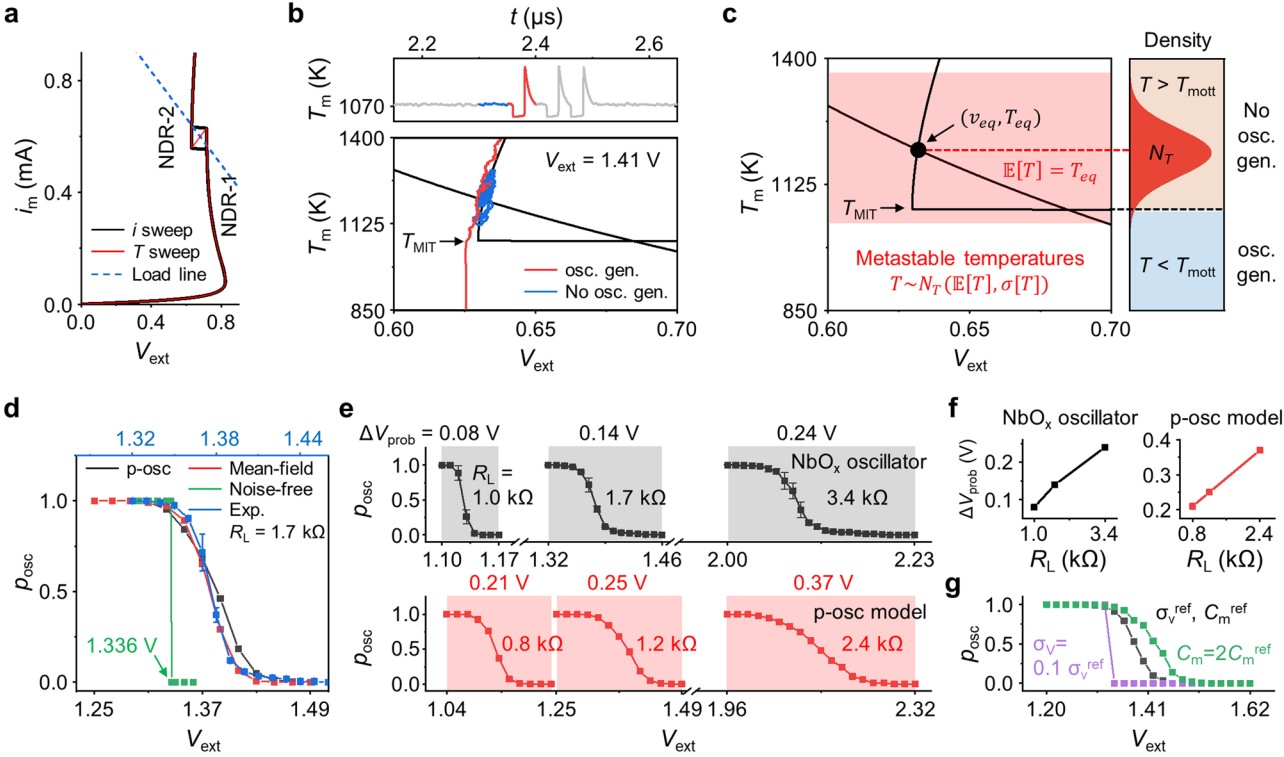

**Fig. 2 | Comprehensive analysis of probabilistic oscillation. a** $I$-$V$ curves of deterministic NbO$_x$ memristor model obtained by current sweep (black) and temperature sweep (red). Load line (blue dashed) at $V_{ext} = 1.41$ V, $R_L = 1.2$ kΩ. **b** $T$-$t$ (upper panel) plot and the corresponding $T$-$V$ plot (lower panel) of the p-osc model highlighting two cases, non-oscillating (blue line) and oscillating (red line). **c** Schematic of the temperature distribution induced by noises in the p-osc model. **d** $p_{osc}$-

$V_{ext}$ of the noise-free model (green), p-osc model (black), mean-field model (red), and experimental result of $R_L = 1.7$ kΩ (blue). Error bars in **d**, **e** show standard deviation. **e**, **f** Experimental and simulation results on voltage range ($\Delta V_{prob}$) of the $p_{osc}$-$V_{ext}$ plots with various $R_L$. **g** The $p_{osc}$-$V_{ext}$ plots with varying voltage noise amplitude ($\sigma_v$) and electrical capacitance ($C_m$).

phase transition between insulating and metallic phases at $T_{MIT}$ (1070 K).

Under the given $R_L$ and $V_{ext}$ conditions, the operating point was formed in the on state above NDR-2, so no oscillation occurs as such. However, when noises were involved, oscillation could occur, and probabilistic oscillation could be understood through the p-osc model as follows. We introduced a time-dependent OU process since white noises were involved in the thermal and electrical conditions of our device (Supplementary Fig. S4). Then, Eqs. 2 and 4 can be replaced by Eqs. 5 and 6.

$$dT(t) = \frac{1}{R_{th}C_{th}}\left(R_{th}iv(t) + T_{amb} - T(t)\right)dt + \sigma_T W(t) \quad (5)$$

$$dv(t) = \frac{1}{R_L C_m}\left(v_{ext} - R_L i - v(t)\right)dt + \sigma_v W(t) \quad (6)$$

In these equations, the stochastic process can originate from the addition of the stochastic diffusion term, $\sigma dW(t)$, where $W(t)$ denotes the Wiener process, and $\sigma^2$ is the variance of the noise.

The noise caused the equilibrium state to be pushed into a metastable state, and the accumulated behaviors could lead to an oscillating state when they reached a specific threshold[30,33], that is $T_{MIT}$ (Supplementary Information section 6). Figure 2b shows $T$-$t$ plot (upper panel) and $T$-$V$ plot (lower panel) obtained from the p-osc model, with $\sigma_T W(t) = 1$ mK and $\sigma_v W(t) = 0.1$ mV. On both panels, non-oscillating and oscillating cases are highlighted in blue and red, respectively. The noise continuously perturbs the equilibrium state of the oscillator, pushing it toward a metastable state out of the equilibrium state (blue line). When the accumulated result reaches the $T_{MIT}$, MIT occurs, thus resulting in a rapid temperature drop until the device reaches the off state. Then, oscillation spike is generated as it spontaneously turns on and returns to the initial equilibrium.

The p-osc model allowed for the calculation of the $p_{osc}$ from the $T$-$t$ plot and, thus, the $p_{osc}$-$V_{ext}$ plot by collecting $p_{osc}$ at various $V_{ext}$, similar to calculating the $p_{osc}$ from experiments in Fig. 1c (p-osc model-based oscillation data is included in Supplementary Fig. S6). The p-osc model-based oscillation results could reasonably reproduce the experimental data (Fig. 1c, d and Supplementary Fig. S2), confirming the credibility of introducing the OU process. However, obtaining the $p_{osc}$-$V_{ext}$ plot through this method is quite complex, and as a result, it is burdensome to get the $p_{osc}$-$V_{ext}$ plot under various conditions.

To effectively design a pbit, more compact theoretical model for the $p_{osc}$-$V_{ext}$ plot is required. Thus we introduced a mean-field approximation that simplifies the noise behavior by representing it in terms of its mean and standard deviation. Then, the mean ($\mathbb{E}[T]$) and standard deviation ($\sigma[T]$) of the metastable temperatures by the noise are given by Eqs. 7 and 8. (A detailed calculation process is described in Supplementary Information section 8)

$$\mathbb{E}[T] = T_{eq} = R_{th}iv_{eq} + T_{amb} \quad (7)$$

$$\sigma[T] = \sqrt{\frac{R_{th}C_{th}}{2}}\left(\sigma_T + \frac{1}{R_{th}C_{th}} \cdot \left.\frac{dT}{dv}\right|_{v_{eq}} \sigma_v \sqrt{\frac{R_L C_m \Delta t}{2}}\right) \quad (8)$$

Figure 2c plots the normal distribution of metastable temperatures ($N_T$) (right panel) at the given conditions (left panel). The device starts oscillation when the metastable temperatures drop below $T_{MIT}$. Therefore, the oscillation probability is given as a function of the area below the $T_{MIT}$ in the distribution. Figure 2d plots $p_{osc}$-$V_{ext}$ of the noise-free model (green), p-osc model (black), mean-field model (red), and experimental result of $R_L = 1.7$ kΩ (blue, upper $x$ axis) reproducing the sigmoidal curve accurately.

From Eq. 8, the voltage range ($\Delta V_{prob}$) of the sigmoidal $p_{osc}$-$V_{ext}$ curve is modulable by adjusting oscillator circuit parameters. Figure 2e, f shows $\Delta V_{prob}$ with various $R_L$ in experiments with $NbO_x$ oscillator and simulation with p-osc model. As $R_L$ increased, $\Delta V_{prob}$ increased, confirming the tunability of the pbit characteristics. In addition, the dependence of $\Delta V_{prob}$ with the voltage noise amplitude and electrical capacitance is shown in Fig. 2g.

## Probabilistic computing demonstration using $NbO_x$ oscillator as a pbit

Here, we implemented a Boltzmann machine[34] adopting the established p-osc model-based pbits ($NbO_x$ pbit) for solving NP-hard problems through simulations. Figure 3a schematically illustrates the memristive Boltzmann machine (MBM), where $P_i$ refers to $NbO_x$ pbit. Memory and process units are the necessary components for storing the outputs and calculating the next inputs for each pbit, respectively. Also, input and output controllers are shown for applying input voltages and reading the output signals.

Here, one operation cycle of the MBM at the $t$-th iteration comprises the following three steps: (Step 1) The $NbO_x$ pbits ($P_i$) receive voltage inputs ($V_{ext,i}^t$) respectively and return probabilistic digital outputs as an oscillation spike. Process unit converts the presence or absence of a spike to a 1 or 0 ($X_i^t \in \{0,1\}$). At each iteration, the output vector $\mathbf{X}^t(X_1^t, \ldots, X_n^t)$ is obtained, which corresponds to a potential answer of the $t$-th iteration. These output vectors are collected in the memory for the final evaluation after all iterations are completed. (Step 2) The processing unit calculates the Hamiltonian gradient ($D_i^t = -\partial h(\mathbf{X}^t)/\partial X_i$) as following the neuronal dynamics between neurons[34,35]. (Step 3) For obtaining the subsequent input voltages, the $D_i^t$ is linearly transformed to the $V_{ext,i}^{t+1}$ by the following equation[36];

$$V_{ext,i}^{t+1} = aD_i^t + b \quad (9)$$

where $a = \frac{V_{osc,min} - V_{osc,max}}{D_{max} - D_{min}}$, $b = V_{osc,min} - aD_{max}$, in which $V_{osc,max}$ and $V_{osc,min}$ are the maximum and minimum input voltages of the pbit. $D_{max}$ and $D_{min}$ are the positive and negative values of the largest absolute value in the range of $D_i^t$. This operating cycle repeats for $N$ iterations. Consequently, $N$ sets of $\mathbf{X}$s are collected in the memory, and the majority $\mathbf{X}$ is determined as an answer.

Using the designed MBM, we solved a minimum vertex covering (MVC) problem. The objective of the MVC problem is to find a subset of vertices that encompasses at least one endpoint of every edge of the undirected and non-weighted graph[37]. When the graph is non-bipartite, the MVC problem is an NP-hard problem, which lies beyond the capabilities of classical algorithms to solve within polynomial time as the problem size increases[38,39]. For example, in the case of the brute-force algorithm, if a graph consists of $n$ vertices and $m$ edges, it requires $2^n \times n \times m$ computations to find the answer in all possible cases (Algorithm S1).

Figure 3b shows a non-bipartite graph, G(6, 7), with six vertices connected by seven edges. Although this problem falls under the category of NP-hard problems, its size ($n = 6$, $m = 7$) is small enough that the solution can be obtained using a classical brute-force algorithm. The MVCs were $\mathbf{X}(X_1, X_2, X_3, X_4, X_5, X_6) = $ '011001' and '101001' as shown in Fig. 3c. Here, note that $X_i$ represents each vertex and the number 0 or 1 shows which of the two groups that vertex is part of.

Next, we derived a solution using our designed $NbO_x$ pbit-based MBM. We adopted an Ising model approach[3], where each vertex is assigned as a pbit, and the Hamiltonian is defined as Eq. 10.

$$h(\mathbf{X}) = \alpha \sum_{u,v \in E}(1 - X_u)(1 - X_v) + \beta \sum_{v \in V} X_v \quad (10)$$

Here $V$ and $E$ are the edge and vertex set in graph G(V, E). $X_u$ and $X_v$ are binary variables on each vertex $u$ (or $v$), and $\alpha$ and $\beta$ are arbitral

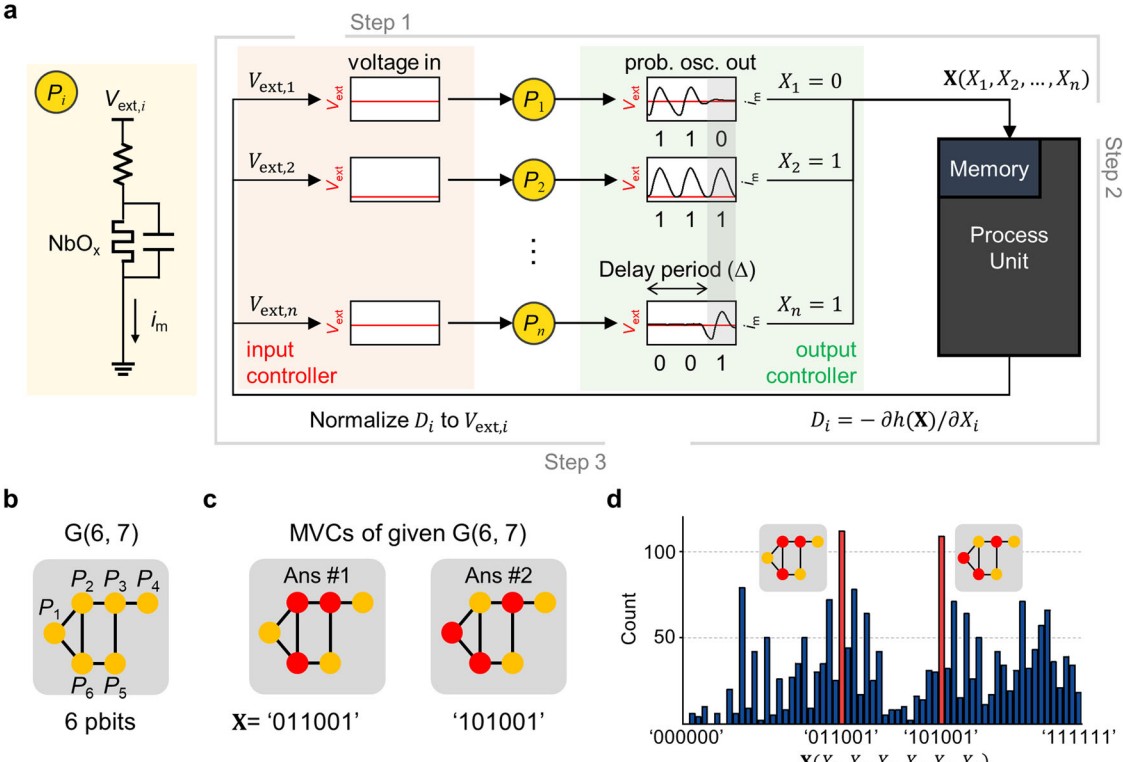

**Fig. 3 | Demonstration of MVC problem solving with the memristive Boltzmann machine (MBM). a** Operation scheme of the constructed MBM. **b** A graph example of G(6, 7) with 6 vertices and 7 edges. **c** Two MVC solutions, $\mathbf{X}(X_1, X_2, X_3, X_4, X_5,$ $X_6$) = '011001' and '101001' obtained by a brute-force algorithm. **d** Histogram of **X** collected over 2000 iterations of the MBM. Most frequent **X**s (red) are same as MVC in **c**.

parameters. Details are described in the Supplementary Information section 9. Figure 3d is a histogram of **X**s collected over 2000 iterations. It shows that the correct answers, **X** = '011001' and 101001', appear most frequently, demonstrating that the MBM solves the given MVC problem.

## Autocorrelation effect of pbit device on p-computing performances

The MBM produces a potential answer at each iteration, and the majority of the answers become the final answer. Therefore, to obtain a reliable final answer, it is necessary to perform a large number of iterations, but this leads to significant time and energy consumption. Consequently, finding the answer within the fewest iterations possible is critical. However, we found that the autocorrelation of pbit significantly decreased the accuracy of the answer and increased the required number of iterations to obtain the answer. We highlight here the autocorrelation issue of pbit outputs and propose a compensating method considering this issue.

Autocorrelation is a serial correlation of time series data. Thus, the autocorrelation involved during the iterative processes can influence the output and change the output probability, deviating from the target probability. This may hinder the efficient search for the solution space. Figure 4a shows probabilistic oscillations at $V_{ext} = 1.37$ V (upper panel) and the corresponding bitstream (lower panel). From the bitstream, the autocorrelation at lag $k$ ($\rho_k$) can be defined as;

$$\rho_k = \frac{\sum_{t=k+1}^{T}(y_t - \bar{y})(y_{t-k} - \bar{y})}{\sum_{t=1}^{T}(y_t - \bar{y})^2} \quad (11)$$

where $y_t$ is the $t$-th bit value, and $\bar{y}$ is the mean of a bitstream. Figure 4b plots the autocorrelations of the bitstream in Fig. 4a as a function of lags from 0 to 20 (Data for all $V_{ext}$ cases are in Supplementary Fig. S8.). At lag 0, the autocorrelation is 1 because it compares to itself, and as

the lag increases, the autocorrelation decreases, suggesting the bits at greater distances are highly independent.

The reason autocorrelation appears can be understood as follows: when the device undergoes oscillation, significant fluctuations occur in temperature and voltage. However, these fluctuations may not fully relax until the subsequent oscillation generates, thereby influencing the initial state of the following oscillation cycle. Therefore, to obtain independent outputs from NbO$_x$ pbits, it is necessary to allow sufficient time between data selection. Thus, we put some delay period ($\Delta$) when determining the pbit output to compensate for autocorrelation. For example, when $\Delta = 2$, the third data of the probabilistic oscillation is chosen for the output. Then, we evaluated the MBM performance with varying $\Delta$. Here, we propose defining the performance as the difference between the occurrence of correct answers ($Y_{cor}$) and the average of the top 5 incorrect answers ($\bar{Y}_{incor}$). This method compares the number of correct answers with the number of major incorrect answers, demonstrating how clear the correct answer is. Detailed explanations are described in Supplementary Information section 12. Figure 4c shows the ($Y_{cor}$ - $\bar{Y}_{incor}$) of one of the answers, **X** = '011001', over 1000 iterations during solving MVC for the ideal case (unautocorrelated, blue) and for the autocorrelated cases without the delay period ($\Delta = 0$, red) and with a delay period of 2 ($\Delta = 2$, violet). When $\Delta = 0$, the ($Y_{cor}$ - $\bar{Y}_{incor}$) was below 0 until more than 600 iterations, indicating that the MBM system gave wrong answers due to the autocorrelation. Whereas when $\Delta = 2$, the ($Y_{cor}$ - $\bar{Y}_{incor}$) was higher than 0, meaning that it could potentially yield the correct answer, which is highly comparable to the ideal case. Figure 4d plots the normalized ($Y_{cor}$ - $\bar{Y}_{incor}$) over 2000 iterations for one of the answers. When $\Delta = 2$, it gives the correct answer steadily after 290 iterations, much smaller than 1310 iterations of $\Delta = 0$, meaning fewer iterations are needed to determine the answer.

Lastly, we have estimated the energy consumption of pbits for the MBM operation in solving the MVC problem. Here, the total energy

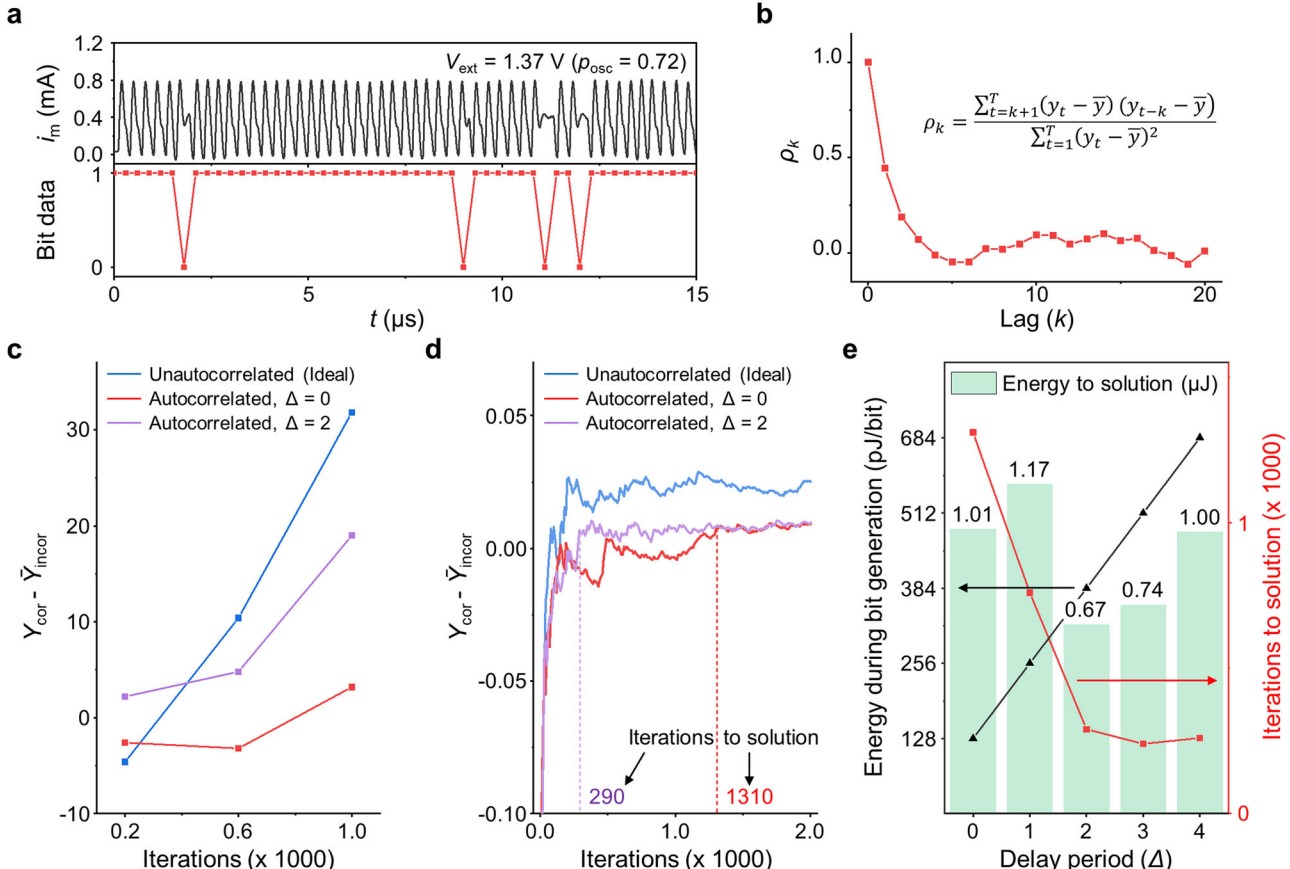

**Fig. 4 | Analysis of autocorrelation issue. a** Probabilistic oscillation plot of NbO$_x$ oscillator at $V_{ext} = 1.37$ V (upper panel) and corresponding bitstream (lower panel). **b** The corresponding autocorrelation ($\rho_k$) of bitstream as a function of lag $k$. **c** Difference between the occurrence of correct answers ($Y_{cor}$) and the average of the top 5 incorrect answers ($\bar{Y}_{incor}$) for uncorrelated case, highly autocorrelated case ($\Delta = 0$), and autocorrelation-relieved case ($\Delta = 2$) as a function of iterations during solving the MVC problem. **d** Normalized ($Y_{cor} - \bar{Y}_{incor}$) value as a function of iterations. **e** Average energy during bit generation (black), iterations to solution (red), and energy to solution (green bars) as a function of $\Delta$.

consumption of pbits (energy to solution) can be approximately defined by (number of pbits, $n$) × (energy during bit generation, $E_{pbit}$) × (number of iterations to solution, $N$). In the energy consumption calculation, we considered only the energy consumption in pbits, and we did not include the energy consumed in memory (storing output vectors) and process units (calculating the Hamiltonian gradient and obtaining the subsequent input voltages from it), as these parts are commonly required in p-computing and are handled by conventional digital computers. Here, $E_{pbit}$ is the energy required for generating the pbit's output. To obtain the exact $E_{pbit}$, it is necessary to count the occurrence of oscillation and non-oscillation cases, which is very complicated. So, we assumed that each case occurs with a 50% probability. Then, $E_{pbit}$ can be set to $128 × (\Delta + 1)$ pJ/bit, where 128 pJ is the average energy per oscillation or non-oscillation and ($\Delta + 1$) is a factor by the delay period. Figure 4e compares the energy efficiency of the MBM as a function of $\Delta$ to get the correct answer for the G(6, 7) problem. Although the energy during bit generation (black) increases proportionally to $\Delta$, the required iteration to the correct answer (red) decreases and converges from $\Delta = 2$ due to sufficient relaxation of autocorrelation. Consequently, the total energy consumption is the lowest at $\Delta = 2$ with 34% less energy consumption than $\Delta = 0$.

To address the autocorrelation issue, we propose allowing sufficient time for state relaxation. However, this inevitably involves time wastage. Therefore, if we can identify the temperature and voltage fluctuation and relaxation characteristics accurately and develop methods to leverage them, the performance of MBMs can be enhanced

without incurring temporal inefficiencies, which should be investigated further.

## Discussion

We developed a new type of pbit using the probabilistic oscillation of the NbO$_x$ memristor that can generate a probabilistic bit in 260 ns with an average energy of 128 pJ/bit in a self-clocking manner. Then, we developed a highly accurate compact model that can fully simulate the experimental results of probabilistic oscillation. We developed a memristive Boltzmann machine composed of NbO$_x$ oscillator-based pbits and solved the graph-based NP-hard problem, validating the feasibility of the proposed pbit. Furthermore, we proposed an auto-correlation issue on the pbit bitstreams and suggested efficient approaches to deal with the issue.

Although this study showed the feasibility of NbO$_x$ oscillator-based pbits, there are still challenges to resolve before it can be practically used. One of the most crucial issues is device-to-device variation. Although our device shows a reliable variation in DC characteristics between cells, any variation may affect the probabilistic oscillation window. Our model suggested that the variation can originate from the difference in noise characteristics injected into each pbit device and from the intrinsic variation of the time-dependent components such as $C_{th}$ and $C_m$. Therefore, future studies will explore related topics such as constant noise supply systems to ensure a consistent environment and low-variation devices with uniform device-to-device parameters to accelerate the development of practical p-computing hardware.

## Methods

### Device fabrication

A TiN/NbO$_x$/TiN-via volatile memristor device was fabricated using the following process. For the 40 nm TiN-via bottom electrode, a planarized substrate containing TiN-vias was prepared from a commercial foundry. A 20 nm-thick NbO$_x$ blanket layer was deposited by reactive sputtering at 100 °C under the mixed gas flow of Ar and O$_2$ (Ar:O$_2$ = 48:2) using an Nb target. Afterward, a 50 nm-thick TiN top electrode and a 20 nm-thick Pt contact electrode were sequentially deposited and patterned by a lift-off process, where the TiN electrode was deposited by reactive sputtering at room temperature using a TiN target, and the Pt electrode was by E-beam evaporation.

### Electrical measurement

All electrical characterizations were performed using a semiconductor analyzer (Keithley 4200A-SCS) and a probe station system. The *I-V* characteristics were obtained in a current sweep using two SMUs (Source Measurement Units). For the self-oscillation characteristics, a 30 μs width of voltage pulses with various levels were applied and measured using a Keithley 4225-PMU (Pulse Measurement Unit) and 4225-RPM (Remote Amplifier/Switch). The *C-V* characteristics were measured using Keithley 4210-CVU (Capacitance Voltage Unit) module.

## Data availability

All the relevant data are available from the corresponding author upon reasonable request.

## Code availability

Simulation results were processed using Python and LTspice software. All the relevant codes are available from the corresponding author upon reasonable request.

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

## Acknowledgements
This research was supported by the National Research Foundation of Korea (NRF) (Grant numbers: RS-2023-00216619, RS-2023-00216992, 2022M3F3A2A01076569, 2022M3I7A4085484, and 2023R1A2C2005159), NNFC (Grant number: 1711160154), and UP program of KAIST (Grant number: N10230061).

## Author contributions
H.R. generated ideas, performed experiments and simulations, and wrote the manuscript. G.K. and J.H.I. provided the initial concept and analyzed its characteristics. Y.L. contributed to the device characteristics analysis during the manuscript revision. H.S. and W.P. fabricated the NbOx device. H.S. and D.H.K. discussed the autocorrelation analysis. K.M.K. supervised this work.

## Competing interests
The authors declare no competing interests.
