## [Peer Review File · Nature Communications]

REVIEWER COMMENTS

Reviewer #1 (Remarks to the Author):

The manuscript, titled "Probabilistic Computing with NbOx Mott Memristor-based Self-oscillatory pbit," explored probabilistic computing in relation to the oscillation of NDR2 in Mott memristor (NbOx). The authors conducted experiments to manipulate the external voltage and control the probabilistic oscillation behavior. They observed that the $V_{ext} - \text{posc}$ curve exhibited a sigmoid shape, satisfying the requirement for pbit. To enhance the existing compact model derived from prior literature, the authors incorporated the Ornstein-Uhlenbeck (OU) process and the stochastic diffusion term. This expanded model was subsequently employed in the field of Probabilistic Computing.

To improve the manuscript and broaden the readership, a minor revision based on following suggestions is recommended.

1. More detailed explanations on Boltzmann machine and MVC problem implemented in this work including the answers of following questions are recommended in Supplementary information.

a. What is D_{max} and D_{min} ? Those need to be defined in the main text.

b. 'We can also solve the problem through~ the energy Hamiltonian as follows'. Did authors do simulations using Ising model approach using Lucas' Hamiltonian only? If so, rephrasing the sentence in line 209~210 is recommended, because it may mislead readers that there are two separate ways to solve MVC. However, when it comes to complicated and big-sized problems, brute-force way will not work and the only way is to use Ising model approach as Figure 3d shows.

c. Since authors tried small-size problem (7-vertices), which might have just a 7x7 matrix, it is recommended to include actual Hamiltonian matrix in Supplementary information for readers.

d. Did authors calculate the energy using the Hamiltonian matrix and a vector of spins, or follow neuronal dynamics of Boltzmann machine (doesn't need to calculate energies)?

e. Boltzmann machine (or Hopfield network) approach of Ising problems requires vector-vector multiplications to update each neuron (spin) state. How was this performed? It is not very clear how to utilize NbO2 oscillators and MVC-solver using Eq.(10).

2. As for Figure 4, it is hard to get the idea on what Y_{cor} and Y_{incor} are. Detailed explanations with an example are recommended to include.

3. Figure 4e, energy to solution is in a.u. However, it is confusing because the authors mention that this is energy per bit (114 pJ/bit) times iterations. Are the results just the number of iterations or including 114 pJ/bit?

4. Grammar errors/typo are recommended to be polished (e.g., page9, 'The goal is the MVC ~')

Reviewer #2 (Remarks to the Author):

In the manuscript authors have introduced “Probabilistic Computing with NbOx Mott Memristor-based Self-oscillatory pbit”. My main points concerning the manuscript are as follows:

1. While the authors argue that the stochasticity observed in the NbO₂ Mott device is merely due to electrical and thermal noise. Due to the composite nature of the fabricated device, the possibility of multiple interacting state variables driving such behavior can not be ignored. In this regard, how well does the addition of the stochastic behavior (as an Ornstein-Uhlenbeck process) model the device behavior? A graphical comparison of the experimentally observed stochastic characteristics obtained from the device with the developed device model could clarify this point.
2. The intrinsic capacitance of the Mott device has a significant role to play in the oscillatory behavior of the device, the authors have not given any quantitative measure of this for the fabricated device.
3. Any firing event in Mott nanodevices is highly correlated with the previous firing event (characterized by relaxation times). Would it be possible to leverage this knowledge of the fabricated device to alleviate the auto-correlation issue?
4. Please mention whether the minimum vertex covering problem was performed with actual devices or through simulation.
5. How well does the designed algorithm for the minimum vertex covering problem perform with respect to any brute-force algorithm performed on a classical computer?

Reviewer #3 (Remarks to the Author):

In this manuscript, H. Rhee et al. present an experimental and modeling-based study of NbOx mott memristor-based oscillators. The primary motivation is to use these oscillators for probabilistic computing, and they do this by controlling the probability of oscillations by external voltages. Probabilistic computing is demonstrated by solving an MVC problem. They also proposed a method to improve the accuracy of the computation by inserting the delay periods.

The main achievement of the present manuscript is identifying the probabilistic oscillation behavior of previously reported NbOx oscillators by fine-tuning the operating parameters. The motivation is timely, and the result is interesting. However, the manuscript as it stands has several aspects that call for further clarifications and improvements, as listed below:

1. NbOx-based self-excited oscillators have already been reported (see Ref. 13 for example), and the application of self-excited oscillators to probabilistic computing is not new. Authors should clarify the significance of their results over previous reports in the manuscript.

2. The authors attributed the metal-insulator transition of NbOx to the mott transition. However, aside from the hysteretic behavior shown in Fig. 1a, sufficient experimental evidence of the mott transition is not shown. Especially how did the authors isolate the mott transition from the lattice effects? Did they consider the possibility of the Peierls transition? Unless other metal-insulator transition mechanisms can be ruled-out, proper rewording should be made to avoid potentially misleading statements.

3. I find the authors' statements "At T_{mott} , the energy of the metallic and insulating phases are equal and separated by an energy barrier, making the mott transition able to offer pbit functionality" (p.3) is also misleading as the thermodynamic process like mott transition is not directly relevant here but only the stability diagram like Figs. S2 c,d is relevant. Otherwise, simply raising the temperature to T_{mott} without currents would be sufficient to observe the oscillations driven by thermal fluctuations.

4. Under large injected currents like in Fig. 1a, the effect of the electrochemical migration and the filament formation should also be considered. Especially around the metal-insulator transition, I expect a spatially inhomogeneous landscape inside the NbOx material due to the phase separation. The authors modeled the NbOx oscillator assuming a spatially homogeneous landscape and claim that the model matches the experiment. Please elaborate on how this assumption is justified. Can the authors devise some experimental approaches to resolve the special inhomogeneity, for example, by measuring the scaling to the device size?

5. I find the electrical noise amplitude of 0.1 mV rather large. What are the sources of the thermal and electrical noises did the authors assume? What makes the authors believe the shot noise (Poisson process) and 1/f noise aren't relevant?

6. The manuscript shows a probabilistic computing demonstration using a NbOx-based Boltzmann machine. However, I don't find any description of the actual implementation of the Boltzmann machine in the manuscript. Such a description is necessary (how many actual NbOx devices are used, how signals are collected and processed, etc.). I am especially interested in how the different NbOx oscillators are synchronized with each other, as the operation scheme described in the manuscript requires such synchronization. Furthermore, the estimation of energy consumption for the MVC task is desirable.

7. Authors should clarify how the device performance of "260 ns with an energy of 114 pJ bit⁻¹" is derived because the energy consumption is expected to vary depending on the computational task.

8. In Figs. 2a and 2b, the simulation is done with the load line inside the NDR-2 hysteresis, and probabilistic oscillation behavior is observed. However, the measurement results in Figs. 1a (red curve) and 1d show the probabilistic oscillations even though the load line is outside the NDR-2 hysteresis (where I expect continuous oscillations rather than probabilistic oscillations). What is the reason for this unexpected observation?

REVIEWER COMMENTS

Reviewer #1 (Remarks to the Author):

The manuscript, titled "Probabilistic Computing with NbOx Mott Memristor-based Self-oscillatory pbit," explored probabilistic computing in relation to the oscillation of NDR2 in Mott memristor (NbOx). The authors conducted experiments to manipulate the external voltage and control the probabilistic oscillation behavior. They observed that the V_{ext} - ρ_{osc} curve exhibited a sigmoid shape, satisfying the requirement for pbit. To enhance the existing compact model derived from prior literature, the authors incorporated the Ornstein-Uhlenbeck (OU) process and the stochastic diffusion term. This expanded model was subsequently employed in the field of Probabilistic Computing.

To improve the manuscript and broaden the readership, a minor revision based on following suggestions is recommended.

>> [Author Response]

Thank you for your thoughtful comments. We have carefully reviewed the comments and prepared one-by-one responses. We hope that our responses will address the reviewer's suggestions appropriately.

1. More detailed explanations on Boltzmann machine and MVC problem implemented in this work including the answers of following questions are recommended in Supplementary information.

>> [Author Response]

Thank you for the feedback. We acknowledge that it wasn't easy for readers to follow the corresponding part. Therefore, in this revision, we have made efforts to improve it. Especially in relation to Boltzmann machine operation and solving the MVC problem (Fig. 3), we made significant revisions to the manuscript and additionally, we added Supplementary Information section 9 to enhance the information.

Supplementary Information section 9

"9. In-depth description of the MBM to solving MVC problem"

As described in the manuscript, in the operation of the Boltzmann machine, the negative derivative of the Hamiltonian ($h(\vec{X})$) for each pbit at the current time step ($D_i^t = -\partial h(\vec{X}^t)/\partial X_i$) is used as the input value for that pbit at the next time step. However, as can be seen from the $h(\vec{X})$ in Equation 10 of main text, D_i^t depends on the absolute size of the $h(\vec{X})$ which is affected by α and β (or additional parameters in other problems⁵) or the way the developed $h(\vec{X})$ is defined. Therefore, when simply using D_i^t as an input value for the next step, an excessively large or small D_i^t may make the stochastic behavior too deterministic, or may not induce significant differences in others. Thus, to mitigate this problem previously, we arithmetically calculated the maximum and minimum values of D_i^t that could be obtained from the initially set $h(\vec{X})$ and normalized D_i^t into the device operating range V_{ext} .

In case of minimum vertex covering (MVC) problem⁵, $h(\vec{X})$ as quadratic unconstrained binary optimization (QUBO) formulation is described as Equation S20 (same as Equation 10 of main text), where $X_v \in \{0, 1\}$.

$$h(\vec{X}) = \alpha \sum_{u,v \in E} (1 - X_u)(1 - X_v) + \beta \sum_{v \in V} X_v \quad (S20)$$

If we rewrite this equation in vector-matrix form, $h(\vec{X})$ is described as Equation S21.

$$h(\vec{X}) = \alpha \vec{X}^T (0.5A - D) \vec{X} + \alpha \mathbb{1}^T (0.5A) \mathbb{1} + \beta \vec{X}^T \vec{X} \quad (\text{S21})$$

$$h(\vec{X}) = X^T \{ \alpha(0.5A - D) + \beta I \} X + 2\alpha \quad (\text{S22})$$

where A is adjacency matrix, D is degree matrix, $\mathbb{1}$ is all-ones matrix of the same size as A and D , and I is identity matrix. If we set each element in $(0.5A - D)$ as P_{ij} , Equation S22 is describes as below:

$$h(\vec{X}) = \vec{X}^T \begin{pmatrix} \alpha P_{11} + \beta & \alpha P_{12} & \cdots & \alpha P_{1n} \\ \alpha P_{21} & \alpha P_{22} + \beta & \cdots & \alpha P_{2n} \\ \vdots & \vdots & \ddots & \vdots \\ \alpha P_{n1} & \alpha P_{n2} & \cdots & \alpha P_{nn} + \beta \end{pmatrix} \vec{X} + 2\alpha \quad (\text{S23})$$

Due the symmetric property of adjacency matrix A ,

$$h(\vec{X}) = \vec{X}^T \begin{pmatrix} \alpha P_{11} + \beta & \alpha P_{12} & \cdots & \alpha P_{1n} \\ \alpha P_{12} & \alpha P_{22} + \beta & \cdots & \alpha P_{2n} \\ \vdots & \vdots & \ddots & \vdots \\ \alpha P_{1n} & \alpha P_{2n} & \cdots & \alpha P_{nn} + \beta \end{pmatrix} \vec{X} + 2\alpha = \vec{X}^T Q \vec{X} + 2\alpha \quad (\text{S24})$$

While calculation of $h(\vec{X})$, $X_v^2 = X_v$ due to $X_v \in \{0,1\}$. Thus, $\partial h(\vec{X}^t) / \partial X_i$ can be simply described as below equation, which aligns with neuronal dynamics between neurons⁶.

$$\frac{\partial h(\vec{X})}{\partial X_i} = Q_{ii} + \sum_{j=1, j \neq i}^n 2Q_{ij} X_j \quad (\text{S25})$$

The off-diagonal elements of Q is equal to corresponding element of $0.5A$ and the adjacency matrix A has only positive value. Thus, the input value of the i^{th} pbit for the next time step can have a range of values as below:

$$-\frac{\partial h(\vec{X})}{\partial X_i} \in \left[-Q_{ii}, -\left(Q_{ii} + \sum_{j=1, j \neq i}^n 2Q_{ij} \right) \right] \quad (\text{S26})$$

So, from the final Equation S26, we can get n sets of maximum and minimum values of $-\frac{\partial h(\vec{X})}{\partial X_i}$ for n pbit systems. We then take the absolute values of all n sets and designated the largest absolute value among the $2n$ values as D_{\max} and $D_{\min} = -D_{\max}$.

In case of graph G(6, 7) in Fig. 3b, the A, D , and Q matrix is defined as below.

$$A = \begin{pmatrix} 0 & 1 & 0 & 0 & 0 & 1 \\ 1 & 0 & 1 & 0 & 0 & 1 \\ 0 & 1 & 0 & 1 & 1 & 0 \\ 0 & 0 & 1 & 0 & 0 & 0 \\ 0 & 0 & 1 & 0 & 0 & 1 \\ 1 & 1 & 0 & 0 & 1 & 0 \end{pmatrix}, D = \begin{pmatrix} 2 & 0 & 0 & 0 & 0 & 0 \\ 0 & 3 & 0 & 0 & 0 & 0 \\ 0 & 0 & 3 & 0 & 0 & 0 \\ 0 & 0 & 0 & 1 & 0 & 0 \\ 0 & 0 & 0 & 0 & 2 & 0 \\ 0 & 0 & 0 & 0 & 0 & 3 \end{pmatrix}, \quad (\text{S27})$$

$$Q = \begin{pmatrix} \beta - 2\alpha & 0.5\alpha & 0 & 0 & 0 & 0.5\alpha \\ 0.5\alpha & \beta - 3\alpha & 0.5\alpha & 0 & 0 & 0.5\alpha \\ 0 & 0.5\alpha & \beta - 3\alpha & 0.5\alpha & 0.5\alpha & 0 \\ 0 & 0 & 0.5\alpha & \beta - \alpha & 0 & 0 \\ 0 & 0 & 0.5\alpha & 0 & \beta - 2\alpha & 0.5\alpha \\ 0.5\alpha & 0.5\alpha & 0 & 0 & 0.5\alpha & \beta - 3\alpha \end{pmatrix}$$

Then, the minimum, maximum value set for each pbit can be derived as below equations. We have set

$\alpha = 2$ and $\beta = 1$ in this research.

$$-\frac{\partial h(\vec{X})}{\partial X_1} \in [-1,3] \quad (\text{S28})$$

$$-\frac{\partial h(\vec{X})}{\partial X_2} \in [-1,5] \quad (\text{S29})$$

$$-\frac{\partial h(\vec{X})}{\partial X_3} \in [-1,5] \quad (\text{S30})$$

$$-\frac{\partial h(\vec{X})}{\partial X_4} \in [-1,1] \quad (\text{S31})$$

$$-\frac{\partial h(\vec{X})}{\partial X_5} \in [-1,3] \quad (\text{S32})$$

$$-\frac{\partial h(\vec{X})}{\partial X_6} \in [-1,5] \quad (\text{S33})$$

Thus, D_{\max} and D_{\min} are designated as 5 and -5 in this case.”

[R1_#1-a] a. What is Dmax and Dmin? Those need to be defined in the main text.

>> **[Author Response]**

Thank you for asking for more clarification on this. They are the maximum and minimum D_i^t values. We added the definition of D_{\max} and D_{\min} in page 9 in main text as follows and described in Equation S26 in Supplementary Information section 9.

Page 9 in main text

“... , where $a = \frac{V_{\text{osc,min}} - V_{\text{osc,max}}}{D_{\max} - D_{\min}}$, $b = V_{\text{osc,min}} - aD_{\max}$, in which $V_{\text{osc,max}}$ and $V_{\text{osc,min}}$ are the maximum and minimum input voltages of the pbit. D_{\max} and D_{\min} are the positive and negative values of the largest absolute value in the range of D_i^t”

Equation S26 in Supplementary Information section 9

“... Thus, the input value of the i^{th} pbit for the next time step can have a range of values as below:

$$-\frac{\partial h(\vec{X})}{\partial X_i} \in \left[-Q_{ii}, -Q_{ii} + \sum_{j=1, j \neq i}^n 2Q_{ij} \right] \quad (\text{S26})$$

So, from the final Equation S26, we can get n sets of maximum and minimum values of $-\frac{\partial h(\vec{X})}{\partial X_i}$ for n pbit systems. We then take the absolute values of all n sets and designated the largest absolute value among the $2n$ values as D_{\max} and $D_{\min} = -D_{\max}$.”

[R1_#1-b] b. ‘We can also solve the problem through~ the energy Hamiltonian as follows’. Did authors

do simulations using Ising model approach using Lucas' Hamiltonian only? If so, rephrasing the sentence in line 209~210 is recommended, because it may mislead readers that there are two separate ways to solve MVC. However, when it comes to complicated and big-sized problems, brute-force way will not work and the only way is to use Ising model approach as Figure 3d shows.

>> [Author Response]

Thank you for precisely pointing out the confusing parts in our paper. We comprehensively rewrote this part in pages 9-10 in main text to make it more clearer. As the reviewer correctly indicated, for small sizes, the brute-force algorithm can be applied, but for large or complex graphs, only the Ising model approach is viable to solve the problem.

Pages 9-10 in main text

“Figure 3b shows a non-bipartite graph, $G(6, 7)$, with six vertices connected by seven edges. Although this problem falls under the category of NP-hard problems, its size ($n = 6, m = 7$) is small enough that the solution can be obtained using a classical brute-force algorithm. The MVCs were $\vec{X}(X_1, X_2, X_3, X_4, X_5, X_6) = '011001'$ and $'101001'$ as shown in Figure 3c. Here, note that X_n represents each vertex and the number 0 or 1 shows which of the two groups that vertex is part of.

Next, we derived a solution using our designed NbO_x pbit-based MBM. We adopted an Ising model approach³, where each vertex is assigned as a pbit, and the Hamiltonian is defined as Equation 10. ...”

[R1_#1-c] c. Since authors tried small-size problem (7-vertices), which might have just a 7x7 matrix, it is recommended to include actual Hamiltonian matrix in Supplementary information for readers.

>> [Author Response]

We appreciate the comment. We have included this matrix in Equations S22 and S27 in Supplementary Information section 9 ($\alpha = 2$ and $\beta = 1$), and have also addressed it in page 10 in main text as follows.

Equations S22 and S27 in Supplementary Information section 9

“

$$h(\vec{X}) = X^T \{ \alpha(0.5A - D) + \beta I \} X + 2\alpha \quad (\text{S22})$$

$$A = \begin{pmatrix} 0 & 1 & 0 & 0 & 0 & 1 \\ 1 & 0 & 1 & 0 & 0 & 1 \\ 0 & 1 & 0 & 1 & 1 & 0 \\ 0 & 0 & 1 & 0 & 0 & 0 \\ 0 & 0 & 1 & 0 & 0 & 1 \\ 1 & 1 & 0 & 0 & 1 & 0 \end{pmatrix}, D = \begin{pmatrix} 2 & 0 & 0 & 0 & 0 & 0 \\ 0 & 3 & 0 & 0 & 0 & 0 \\ 0 & 0 & 3 & 0 & 0 & 0 \\ 0 & 0 & 0 & 1 & 0 & 0 \\ 0 & 0 & 0 & 0 & 2 & 0 \\ 0 & 0 & 0 & 0 & 0 & 3 \end{pmatrix},$$

(S27)

$$Q = \begin{pmatrix} \beta - 2\alpha & 0.5\alpha & 0 & 0 & 0 & 0.5\alpha \\ 0.5\alpha & \beta - 3\alpha & 0.5\alpha & 0 & 0 & 0.5\alpha \\ 0 & 0.5\alpha & \beta - 3\alpha & 0.5\alpha & 0.5\alpha & 0 \\ 0 & 0 & 0.5\alpha & \beta - \alpha & 0 & 0 \\ 0 & 0 & 0.5\alpha & 0 & \beta - 2\alpha & 0.5\alpha \\ 0.5\alpha & 0.5\alpha & 0 & 0 & 0.5\alpha & \beta - 3\alpha \end{pmatrix}$$

”

Page 10 in main text

“... Details are described in the Supplementary Information section 9. ...”

[R1_#1-d] d. Did authors calculate the energy using the Hamiltonian matrix and a vector of spins, or follow neuronal dynamics of Boltzmann machine (doesn't need to calculate energies)?

>> [Author Response]

Thank you for the insightful question. In our study, we use a memristive Boltzmann machine (MBM), where the operational procedure aligns with the neuronal dynamics between neurons rather than explicit energy calculations. Therefore, there is no need to actually calculate the Hamiltonian. The statement about calculating the Hamiltonian on page 9 in main text is unnecessary so we deleted it as follows.

Page 9 in main text

“... (Step 2) The processing unit calculates ~~the Hamiltonian ($h(\vec{X}^t)$) and the Hamiltonian gradient ($D_i^t = -\partial h(\vec{X}^t)/\partial X_i$)~~ as following the neuronal dynamics between neurons^{32,33}. ...”

Moreover, we clarified it in Equation S25 in Supplementary Information section 9.

Equation S25 in Supplementary Information section 9

“... Thus, $\frac{\partial h(\vec{X}^t)}{\partial X_i}$ can be simply described as below equation, which aligns with neuronal dynamics between neurons⁶.

$$\frac{\partial h(\vec{X})}{\partial X_i} = Q_{ii} + \sum_{j=1, j \neq i}^n 2Q_{ij} X_j \quad (\text{S25})$$

The off-diagonal elements of Q is equal to corresponding element of $0.5A$ and the adjacency matrix A has only positive value. Thus, the input value of the i^{th} pbit for the next time step can have a range of values as below: ...”

To be Specific, the next state of each pbit is defined by the negative derivative of the Hamiltonian. This process makes the pbit’s input at the subsequent timestep towards a state that lowers the system's energy. Consequently, explicit energy calculation at each timestep is becomes unnecessary.

In essence, using the negative derivative of the Hamiltonian aligns with the concept of neuronal dynamics, where each neuron updates its state based on the weighted sum of the states of its neighboring neurons^{1,2}. This comes from the Ising model employed by our Boltzmann machine system, wherein neurons interact pairwise with specified weights. Process within Equations S20-S27 explain this clearly, demonstrating that these processes are functionally identical.

[R1_#1-e] e. Boltzmann machine (or Hopfield network) approach of Ising problems requires vector-vector multiplications to update each neuron (spin) state. How was this performed? It is not very clear how to utilize NbO₂ oscillators and MVC-solver using Eq.(10).

>> [Author Response]

We appreciate your detailed comment. As previously mentioned in comment [R1_#1-d], our MBM utilizes neuronal dynamics during problem-solving operation (updating neuron states), i.e., searching for the lowest energy. The Hamiltonian is defined as the vector-vector multiplication of the output states of each NbO_x pbit, as outlined in Equation 10 in main text, or in its equivalent representation, Equation S22 in Supplementary Information section 9.

Equation 10 in main text

“

$$h(\vec{X}) = \alpha \sum_{u,v \in E} (1 - X_u)(1 - X_v) + \beta \sum_{v \in V} X_v \quad (10)$$

”

Equation S22 in Supplementary Information section 9

“

$$h(\vec{X}) = X^T\{\alpha(0.5A - D) + \beta I\}X + 2\alpha \quad (\text{S22})$$

”

However, it is crucial to note that the actual operation conducted by the MVC-solver is not a vector-vector multiplication, but rather it is represented as a linear equation, as indicated in Equation S25 in Supplementary Information section 9 due to using neuronal dynamics^{1,2}.

Equation S25 in Supplementary Information section 9

“

$$\frac{\partial h(\vec{X})}{\partial X_i} = Q_{ii} + \sum_{j=1, j \neq i}^n 2Q_{ij} X_j \quad (\text{S25})$$

”

We constructed a Boltzmann machine system through simulation, using the NbO_x pbit model. During this process, the MVC-solver updated the state of each pbit synchronously, sequentially updating each pbit at every timestep. The updating process followed a fixed sequence: 1 - 2 - 3 - ... - 6. Upon completion of the six updates, the iteration ended, and the next iteration commenced, using the end state of the previous iteration as its starting state.

To provide an example, in a 6 pbit MBM system where the delay period was set to 2, the MVC-solver applied an input voltage to each pbit to initiate oscillation with a probability of 0.5 when the first iteration began. Then, the MVC-solver read the third oscillation states of each pbit in the first timestep (corresponding to the first pbit update step), given that the delay period was set to 2. Subsequently, the MVC-solver calculated the negative derivative of the energy corresponding to the first pbit in accordance with Equation S25. After the normalization step to fit the calculated input into NbO_x pbit operation voltage range, this final input was re-input to the corresponding NbO_x pbit. This process was conducted for the remaining pbits during iteration process.

[R1_#2] 2. As for Figure 4, it is hard to get the idea on what Y_{cor} and Y_{incor} are. Detailed explanations with an example are recommended to include.

>> **[Author Response]**

Thank you for your insightful comment. We understand that the concepts of Y_{cor} and \bar{Y}_{incor} in Fig. 4 may require additional clarification. We added more explanation on page 11 in main text as follows.

Page 11 in main text

“... Here, we propose defining the performance as the difference between the occurrence of correct answers (Y_{cor}) and the average of the top 5 incorrect answers (\$\bar{Y}_{incor}\$ ). This method compares the number of correct answers with the number of major incorrect answers, demonstrating how clear the correct

answer is. Detailed explanations are described in Supplementary Information section 12. ...”

Also, we have elaborated on them in Supplementary Information section 12, which includes an energy heatmap for all possible \vec{X} based on the defined energy Hamiltonian.

Supplementary Information section 12

“12. Energy heatmap and detailed explanation of \$Y_{cor}\$ and \$\bar{Y}_{incor}\$ in the MBM system

Fig. S9. a Given graph $G(6, 7)$ and its MVCs. **b, c** Energy maps for every states (\vec{X}) based on defined Hamiltonian (Equation 10 in main text) to solving MVC problem.

Fig. S9a shows the minimal vertex coverings (MVCs) of the $G(6, 7)$ graph presented in the main text. Given the relatively small scale of the problem, it is feasible to identify the correct ($\vec{X} = '011001'$ and $'101001'$) and incorrect solutions using a brute-force algorithm.

The energy maps according to the Hamiltonian (Equation 10) is shown in Figs. S9b-c. This is implemented in the MBM, employing the Ising model approach⁵. The energy map reveals that the MBM system assigns global minimum energy states to $\vec{X} = '011001'$ and $'101001'$, the same as the ground truth from the brute-force algorithm. There are also local minimum states, such as $'010111'$, $'011011'$, $'011101'$, $'101011'$, $'101101'$, $'111110'$, $'111001'$, and $'111010'$, which are energetically favorable compared to their immediate surroundings but hold higher energy than the global minimum states.

Fig. S10. **a** Energy map of MVC problem for given graph $G(6, 7)$. MBM simulation results for every \vec{X} states at 1,000 iterations in case of **b** un-autocorrelated, **c** autocorrelated, no delay period (Δ), **d** autocorrelated, $\Delta = 2$.

In accordance with the Boltzmann law, global minimum states appear most frequently, followed by the local minimum states. Thus, we anticipated that \bar{Y}_{incor} , which represents the average occurrence of the five most frequently appearing non-correct cases, would primarily consist of these local minimum states. Figs. S10b-d validate this assumption by displaying the MBM simulation results for every configuration \vec{X} in both the presence and absence of autocorrelation.

Autocorrelation-induced non-independent probabilistic behavior hinders the MBM from effectively exploring the solution space represented by the energy map, leading to an increase in occurrence of local minimum states. Consequently, we observe this effect as a rise in \bar{Y}_{incor} in Figs. S10c-d as well as Figs. 4c-d in the main text. More specifically, as shown in Fig. S10c, without the inclusion of a delay period, the '011101' configuration becomes most prominent in the presence of autocorrelation. This highlights the risk that autocorrelation could cause a local minimum state to appear dominantly within a small number of iterations, thereby deviating from the ground truth."

[R1_#3] 3. Figure 4e, energy to solution is in a.u. However, it is confusing because the authors mention that this is energy per bit (114 pJ/bit) times iterations. Are the results just the number of iterations or including 114 pJ/bit?

>> [Author Response]

Thank you for the valuable feedback. In this revision, we have calculated the energy and updated the y-axis in Fig. 4e in main text to represent energy values as follows.

Fig. 4e in main text

Revised y-axis title: “Energy during bit generation”

Furthermore, we added the energy calculation method in page 12 in main text as follows.

Page 12 in main text

“Lastly, we have estimated the energy consumption of pbits for the MBM operation in solving the MVC problem. Here, the total energy consumption of pbits (energy to solution) can be approximately defined by (number of pbits, n) \times (energy during bit generation, E_{pbit}) \times (number of iterations to solution, N). Here, E_{pbit} is the energy required for generating the pbit’s output. To obtain the exact E_{pbit} , it is necessary to count the occurrence of oscillation and non-oscillation cases, which is very complicated. So, we assumed that each case occurs with a 50% probability. Then, E_{pbit} can be set to $128 \times (\Delta + 1)$ pJ/bit, where 128 pJ is the average energy per oscillation or non-oscillation and $(\Delta + 1)$ is a factor by the delay period. ...”

[R1_#4] 4. Grammar errors/typo are recommended to be polished (e.g., page9, ‘The goal is the MVC ~’)

>> [Author Response]

Thank you for pointing out the grammatical errors and typos in the manuscript. Following your suggestion, we have thoroughly reviewed and polished the text, including the specific error you noted on page 9, to ensure a smoother reading experience for all. We highly value your contribution to the quality of our work.

Reviewer #2 (Remarks to the Author):

In the manuscript authors have introduced “Probabilistic Computing with NbOx Mott Memristor-based Self-oscillatory pbit”. My main points concerning the manuscript are as follows:

>> [Author Response]

We express our sincere gratitude for your careful review and constructive comments on our manuscript. Your expertise in this field has greatly helped us improve our work. In the following, we have provided responses to each of your points individually and hope that these adequately address your concerns.

[R2_#1] 1. While the authors argue that the stochasticity observed in the NbO₂ Mott device is merely due to electrical and thermal noise. Due to the composite nature of the fabricated device, the possibility of multiple interacting state variables driving such behavior can not be ignored. In this regard, how well does the addition of the stochastic behavior (as an Ornstein-Uhlenbeck process) model the device behavior? A graphical comparison of the experimentally observed stochastic characteristics obtained from the device with the developed device model could clarify this point.

>> [Author Response]

Thank you for the insightful comment. In the original manuscript, we collected the oscillation results as a function of the V_{ext} as shown in Supplementary Fig. S2, and we presented the simulation-based oscillation results incorporating the OU process as shown in Supplementary Fig. S6. We acknowledge that the lack of precise mention of these comparison between Fig. 1 in main text (or Supplementary Fig. S2) and Supplementary Fig. S6 made it difficult to ascertain the consistency between the experimental and simulation results, as pointed out by the reviewer.

In this revision, we elaborated on the detailed description on Supplementary Fig. S2 and S6 and added explanation highlighting the concordance between the experimental and simulation results on **pages 4 and 7 in main text** as follows.

Page 4 in main text

“... Each p_{osc} value was an average from 5 datasets recorded for 30 μs each (Supplementary Fig. S2). ...”

Page 7 in main text

“The p-osc model allowed for the calculation of the p_{osc} from the T - t plot and, thus, the $p_{\text{osc}}-V_{\text{ext}}$ plot by collecting p_{osc} at various V_{ext} , similar to calculating the p_{osc} from experiments in Fig. 1c (p-osc model-based oscillation data is included in Supplementary Fig. S6). The p-osc model-based oscillation results could reasonably reproduce the experimental data (Figs. 1c-d and Supplementary Fig. S2), confirming the credibility of introducing the OU process. ...”

In addition, we added experimental $p_{\text{osc}}-V_{\text{ext}}$ plot in Fig. 2d to directly compare the experimental and simulation results. Furthermore, we have added Fig. 2f to enhance the clarity of the comparison between experimental and simulation results. Manuscript in page 8 in main text is modified also.

Fig. 2 in main text

Fig. 2 Comprehensive analysis of probabilistic oscillation. **a** I - V curves of deterministic NbO_x memristor model obtained by current sweep (black) and temperature sweep (red). Load line (blue dashed) at $V_{\text{ext}} = 1.41$ V, $R_L = 1.2$ k Ω . **b** T - t (upper panel) plot and the corresponding T - V plot (lower panel) of the p-osc model highlighting two cases, non-oscillating (blue) and oscillating (red). **c** Schematic of the temperature distribution induced by noises in the p-osc model. **d** p_{osc} - V_{ext} of the noise-free model (green), p-osc model (black), mean-field model (red), and experimental result of $R_L = 1.7$ k Ω (blue). Error bars in **d-e** show standard deviation. **e, f** Experimental and simulation results of the p_{osc} - V_{ext} plots with various R_L . **g** The p_{osc} - V_{ext} plots with varying voltage noise amplitude (σ_V) and electrical capacitance (C_m).

Page 8 in main text

“... **Figure 2d** plots p_{osc} - V_{ext} of the noise-free model (green), p-osc model (black), mean-field model (red), and experimental result of $R_L = 1.7$ k Ω (blue, upper x axis) reproducing the sigmoidal curve accurately.”

“... **Figure 2e-f** shows ΔV_{prob} with various R_L in experiments with NbO_x oscillator and simulation with p-osc model. ...”

[R2_#2] 2. The intrinsic capacitance of the Mott device has a significant role to play in the oscillatory behavior of the device, the authors have not given any quantitative measure of this for the fabricated device.

>> [Author Response]

We appreciate your comment. In oscillation modeling, the device capacitance is highly critical. We already had these results secured although not explicitly mentioned in the main text. We have included these results on page 4 in main text and provided a detailed description in the Supplementary Information section 1.

“... The device capacitance was 6.47 pF (Supplementary Fig. S1) and ...”

Supplementary Information section 1

“1. Internal capacitance measurement result of NbO_x volatile memristor

Fig. S1. a C-V measurement result of NbO_x memristor with diameter of 40 nm. b Capacitances of NbO_x memristors with different diameter (100 nm, 60 nm, 40 nm) before (pristine) and after initial voltage sweep.

Fig. S1a shows that the NbO_x volatile memristor, constructed with a TiN/NbO_x/TiN via (40 nm diameter), has an internal capacitance of 6.47 pF at a DC voltage of 0 V (off state). The capacitance-voltage (C-V) measurement was conducted by using a semiconductor analyzer (Keithley 4200-SCS) with a C-V module. The C-V measurement was set in sweep mode, with a start voltage of -0.1 V, a stop voltage of 0.1 V, a step increment of 0.01 V, an AC frequency of 1 MHz, and an AC drive voltage of 30 mV RMS. In addition, the capacitance measurements of the pristine and after the initial sweep of the device in Fig. S1b show that the device has a constant capacitance in after the initial sweep operation, indicating that the NbO_x memristor has a uniform active core structure.”

[R2_#3] 3. Any firing event in Mott nanodevices is highly correlated with the previous firing event (characterized by relaxation times). Would it be possible to leverage this knowledge of the fabricated device to alleviate the auto-correlation issue?

>> [Author Response]

Thank you for your insightful comment. As the reviewer expected, the autocorrelation appears because the oscillation state is not fully relaxed until the subsequent oscillation, and therefore, the subsequent oscillation is influenced. In our manuscript, we resolved this autocorrelation issue through the delay period, which means that during this delay period, the firing information from the previous state is reset or initialized.

We believe that leveraging the information of the firing event suggested by the reviewer, rather than waiting for the complete relaxation of the firing event, is a highly valuable approach. However, despite much contemplation, finding a suitable solution for this approach does not appear to be an easy task.

Therefore, in this revision, we will enhance the discussion on page 11 in main text regarding the reasons behind the appearance of autocorrelation and the meaning of our proposed solution. Also, we will include a discussion on page 12 in main text that suggests the potential for leveraging the

autocorrelation issue for computing as follows.

Page 11 in main text

“The reason autocorrelation appears can be understood as follows: when the device undergoes oscillation, significant fluctuations occur in temperature and voltage. However, these fluctuations may not fully relax until the subsequent oscillation generates, thereby influencing the initial state of the following oscillation cycle. Therefore, to obtain independent outputs from NbO_x pbits, it is necessary to allow sufficient time between data selection. ...”

Page 12 in main text

“To address the autocorrelation issue, we propose allowing sufficient time for state relaxation. However, this inevitably involves time wastage. Therefore, if we can identify the temperature and voltage fluctuation and relaxation characteristics accurately and develop methods to leverage them, the performance of MBMs can be enhanced without incurring temporal inefficiencies, which should be investigated further.”

[R2_#4] 4. Please mention whether the minimum vertex covering problem was performed with actual devices or through simulation.

>> **[Author Response]**

Thank you for your comments to improve the clarity of the manuscript. Our MBM demonstration was based on simulation adopting the actual device characteristics. To avoid misleads, we clarified it on page 8 in main text as follows.

Page 8 in main text

“Here, we implemented a Boltzmann machine³² adopting the established p-osc model-based pbits (NbO_x pbit) for solving NP-hard problems through simulations. ...”

[R2_#5] 5. How well does the designed algorithm for the minimum vertex covering problem perform with respect to any brute-force algorithm performed on a classical computer?

>> **[Author Response]**

Thank you for your insightful question about the comparative efficiency between our Ising-based Boltzmann machine (BM) and traditional brute-force algorithms for solving NP-hard problems like the minimum vertex cover (MVC). Efficiency is a multi-faceted concept that encompasses not only computational time but also energy consumption, error rates, and other variables.

To clarify the comparison, we have added the brute-force algorithm of MVC problem to the Supplementary Information section 10.

Supplementary Information section 10

“10. Brute-force algorithm for minimum vertex cover problem

The brute-force algorithm for solving the MVC problem proceeds as follows:

- 1. Initialize:** For a graph G composed of n vertices and m edges, create a matrix V of length n .
- 2. Generate subset:** Produce all possible subsets v of V .
- 3. Check for vertex cover:** For each subset v of V , perform the following steps until all subsets have been examined:
 - 3-1. Determine if v is a vertex cover by checking if every edge (u, v) in E is incident to at least one vertex in v .
 - 3-2. If v is a vertex cover, compare its size (the sum of all elements in v) to the size of the current smallest vertex cover.
 - 3-3. Update the minimum vertex cover to v if its size is smaller.
- 4. Return result:** Return the minimum vertex cover.

Its pseudo code is shown below:

Algorithm S1: Brute-force algorithm for minimum vertex cover problem

```
// Input: Graph G with n vertices and m edges
// Output: Minimum vertex cover. min_vertex_cover

// Initialize
min_vertex_cover = {} // Empty set
min_size = INF // Initialize to a value greater than the maximum possible size (INF is infinity)
V = [V1, V2, ..., Vn] // List of vertices

// Generate Subsets
all_subsets = PowerSet(V) // Generate all subsets of V

// Check for Vertex Cover
for each subset v in all_subsets:
    is_vertex_cover = True

    // Check if v is a vertex cover
    for each edge (u, w) in G.edges:
        if u not in v and w not in v:
            is_vertex_cover = False
            break

    // Compare its size to the current smallest vertex cover
    if is_vertex_cover:
        size_v = len(v)

        if size_v < min_size:
            min_size = size_v
            min_vertex_cover = v // Update the minimum vertex cover

// Return Result
return min_vertex_cover
```

”

This approach has a time complexity of $O(2^n \times n \times m)$, making it highly inefficient for large n due to the exponential increase in time and energy requirements.

In contrast, probabilistic algorithms like BM offer a more efficient alternative. These algorithms find approximate solutions by identifying the lowest energy states in a corresponding energy landscape. While the concept of time complexity is less straightforward for these stochastic systems, they often outperform brute-force algorithms in terms of speed and energy efficiency. In the case of BM, we derive the solution from the distribution of results across multiple iterations. Therefore, empirical metrics such as convergence time, success rate, variance, and scaling behavior are more relevant for performance evaluation.

However, the complexity and difficulty of graph-based combinatorial optimization problems like MVC can vary significantly depending on the specific structure of the graph. This makes it challenging to set a standard for quantitative comparison. Furthermore, the diversity of connectivity of graph becomes increasingly complex as the size of the graph grows, and the results from BM are theoretically strongly dependent on the complexity of the energy surface. Therefore, it's also difficult to compare the performance of the two algorithms simply by increasing the size of the problem.

Nevertheless, it is possible to make a rough comparison between BM and brute-force algorithms by leveraging the statistical nature of BM. For instance, consider an MVC problem with $n = 30$ and $m = 435$ (assuming half of all possible edges are connected). A brute-force algorithm would require approximately $2^{30} \times 30 \times 435 = 1.3 \times 10^{13}$ computations. In contrast, if BM uses k iterations to statistically derive an approximate solution, even collecting ten million samples would be orders of magnitude more efficient than the brute-force approach.

This efficiency gap is further supported by the Boltzmann law, which governs the probability distribution of states in BM. For a brute-force algorithm, we don't know if the state we're currently examining is close to the correct answer or not, so we have to cover all cases. However, for BM, even with a smaller sample size, the most frequently occurring state is likely to be either the correct solution or a close approximation due to the governing Boltzmann law.

Additionally, BM allows for another level of fine-tuning through the adjustment of the Hamiltonian's coefficients. In the Ising model, the Hamiltonian function describes the energy of a particular configuration of spins (or in our case, vertices). It usually includes terms that represent interactions between neighboring spins, and the coefficients of these terms dictate the strength of these interactions. By carefully adjusting these coefficients, we can influence the system's behavior in various ways. For example, we might speed up the convergence to a solution, improve the accuracy of the found solution, or even enable the system to explore a broader range of potential solutions more effectively.

In conclusion, while it's challenging to make a direct comparison between our MBM algorithm and brute-force methods due to inherent complexities and variables, our empirical evidence strongly suggests that MBM offers a more efficient and practical solution for solving MVC problems, especially as the problem size increases. Furthermore, we have elaborated on this in the page 9 in main text as follows.

Page 9 in main text

“... For example, in the case of the brute-force algorithm, if a graph consists of \$n\$ vertices and \$m\$ edges, it requires \$2^n \times n \times m\$ computations to find the answer in all possible cases (Algorithm S1). ...”

Reviewer #3 (Remarks to the Author):

In this manuscript, H. Rhee et al. present an experimental and modeling-based study of NbO_x mott memristor-based oscillators. The primary motivation is to use these oscillators for probabilistic computing, and they do this by controlling the probability of oscillations by external voltages. Probabilistic computing is demonstrated by solving an MVC problem. They also proposed a method to improve the accuracy of the computation by inserting the delay periods.

The main achievement of the present manuscript is identifying the probabilistic oscillation behavior of previously reported NbO_x oscillators by fine-tuning the operating parameters. The motivation is timely, and the result is interesting. However, the manuscript as it stands has several aspects that call for further clarifications and improvements, as listed below:

>> [Author Response]

We greatly appreciate acknowledging the importance of the probabilistic computing research and giving constructive feedbacks on our study. Your insights have certainly helped us refine our work. We have provided detailed responses to each of your comments in the subsequent sections. We believe these responses and the corresponding revisions to our manuscript will address your concerns effectively.

[R3_#1] 1. NbO_x-based self-excited oscillators have already been reported (see Ref. 13 for example), and the application of self-excited oscillators to probabilistic computing is not new. Authors should clarify the significance of their results over previous reports in the manuscript.

>> [Author Response]

I appreciate your feedback regarding the aspect of novelty in our research that we might not have emphasized adequately. While the stochastic oscillation characteristics of NbO_x have been widely discussed, the application of these characteristics to pbits is indeed a novel contribution introduced by our research for the first time.

The initial paper on pbits utilized magnetic tunneling junction (MTJ) device, where the MTJ itself wasn't new³⁻⁶. As such, while the concept of NbO_x-based oscillators might not be entirely novel, finding its stochasticity and proposing their utilization as pbits is indeed a valuable contribution. Moreover, our pbit differs from conventional pbits in that it generates a self-sustaining bitstream without the need for bit-generating signal pulses.

We emphasized this on page 3 in main text as follows.

Page 3 in main text

“Thus, by combining the probabilistic behavior of bi-stability with the oscillation characteristics, it is possible to obtain the probabilistic oscillation, which can be potentially used as a new type of pbits. Moreover, differing from conventional pbits, such oscillator-based pbits can generate a self-sustaining bitstream without bit-generating signal pulses. ...”

Furthermore, we have developed a reliable model for stochastic oscillation based on the Ornstein-Uhlenbeck (OU) process, and we have also proposed the utilization of pbits constructing MBM. All of these topics indeed carry a sense of novelty and innovation.

[R3_#2] 2. The authors attributed the metal-insulator transition of NbO_x to the mott transition. However, aside from the hysteretic behavior shown in Fig. 1a, sufficient experimental evidence of the mott

transition is not shown. Especially how did the authors isolate the mott transition from the lattice effects? Did they consider the possibility of the Peierls transition? Unless other metal-insulator transition mechanisms can be ruled-out, proper rewording should be made to avoid potentially misleading statements.

>> [Author Response]

Thank you for your comments, which have improved the clarity of the manuscript. The mott transition of NbO₂ is a widely accepted theory in the academic community⁷⁻¹⁵, while the Peierls transition is also a explainable theory. The core of our research is not so much about what the transition mechanism is but rather the potential applications that can be derived from it. Therefore, we revised "mott transition" to "metal-insulator transition" over the revise manuscript including the title not to cause any unnecessary confusion.

Title

Revised title: "Probabilistic Computing with NbO_x Metal-Insulator Transition -based Self-oscillatory pbit"

[R3_#3] 3. I find the authors' statements "At T_{mott}, the energy of the metallic and insulating phases are equal and separated by an energy barrier, making the mott transition able to offer pbit functionality" (p.3) is also misleading as the thermodynamic process like mott transition is not directly relevant here but only the stability diagram like Figs. S2 c,d is relevant. Otherwise, simply raising the temperature to T_{mott} without currents would be sufficient to observe the oscillations driven by thermal fluctuations.

>> [Author Response]

Thank you for your insightful comment. We agree that our statement might mislead readers. We revised the corresponding part on page 3 in main text is as follows:

Page 3 in main text

"... The MIT at the T_{MIT} can be easily disturbed by slight irregularities, making the device offer pbit functionality. ..."

Furthermore, as the reviewer mentioned, simply heating the system to approximately 1070K, T_{MIT}, would have induced a phase transition in NbO_x, consistent with our goals. However, we controlled the state of NbO_x through external voltage regulation instead, for two main concerns:

1. Simplicity of the device operating system. Applying heat directly to the device would require a more complex system than simply applying voltage and could cause destruction or deformation to entire system due to high temperatures, including potentially affecting the measurement process. Considering the future fabrication of NbO_x pbit-based probabilistic computing hardware through FPGA, this was not an appropriate method, so we adopted the approach of controlling the state of the device through external voltage.

2. The detection aspect of the transition. In the method of inducing a transition by raising the system's temperature, distinguishing the transition could be done by detecting voltage changes at a constant current when considering the *I-V* curve. However, the voltage difference detected between insulating and metallic phases is less than 0.1 V, making it very challenging considering the thermal effects by heating element in circuit. Therefore, we ensured that the transition induced oscillation, allowing us to clearly detect the results induced by the phase transition of NbO_x.

Consequently, we could definitively detect the probabilistic metal-to-insulator transition through probabilistic oscillating spikes whose probability can be controlled by external voltage. We modified the main text to emphasize our core idea further, and the revised part is the same as the response in [R3_#1].

[R3_#4] 4. Under large injected currents like in Fig. 1a, the effect of the electrochemical migration and the filament formation should also be considered. Especially around the metal-insulator transition, I expect a spatially inhomogeneous landscape inside the NbO_x material due to the phase separation. The authors modeled the NbO_x oscillator assuming a spatially homogeneous landscape and claim that the model matches the experiment. Please elaborate on how this assumption is justified. Can the authors devise some experimental approaches to resolve the special inhomogeneity, for example, by measuring the scaling to the device size?

>> **[Author Response]**

We thank for careful examination of our work. As the reviewer pointed out, it is indeed reasonable to anticipate that the internal active region and its surroundings in the NbO_x have inhomogeneous (continuous) landscape, rather than being composed of homogeneous regions.

However, core-shell models that simplify this inhomogeneity are well known, and reported papers have shown that the core-shell modeling approach simulates the properties of real devices well, despite the fact that devices are predicted to be internally inhomogeneous^{9,11,16-20}. We acknowledge that we have overlooked the detailed information of the internal structure of NbO_x in this paper, as we have primarily focused on modeling experimental phenomena, but we are confident that the modeling approach we have used is reasonable enough to simulate NbO_x devices.

Thus, in the revised manuscript, we have expanded the description of the modeling we used as follows.

Page 4 in main text

“... and it exhibited repeatable double negative differential resistance (NDR) behaviors at the localized region following core-shell structure^{15-18,20-22}. ...”

In addition, the NbO_x device may show the non-volatile switching behavior accompanied by the migration of the oxygen vacancies at a higher current. This has been also widely reported elsewhere. In our device, we could observe a repeatable volatile switching behaviors at the given current level, suggesting the current level is still lower to induce the oxygen vacancy migration. We also elaborated on it in the revised manuscript as follows.

Page 4 in main text

“... During the DC *I-V* sweep, the operating current was sufficiently low not to involve non-volatile resistance changes caused by the electrochemical reaction, such as oxygen vacancy formation^{23,24}. ...”

[R3_#5] 5. I find the electrical noise amplitude of 0.1 mV rather large. What are the sources of the thermal and electrical noises did the authors assume? What makes the authors believe the shot noise (Poisson process) and 1/f noise aren't relevant?

>> **[Author Response]**

We thank the reviewer for the insightful inquiry into the sources of noise in our system. In electrical circuits, white noise has been known to be generated through thermal noise, which results from the excitation of electrons by heat, or shot noise, which results from the discretization of electrons at junctions²¹. However, it is difficult to pinpoint a specific source because the environment in which the device is measured includes noise from the surrounding environment, noise from the measurement system, and many other sources.

Therefore, we investigated the types of electrical noise in our system as described in Supporting Information section 5.

Supporting Information section 5

5. Electrical noise analysis on NbO_x oscillator

Fig. S4. a Circuit diagram of NbO_x oscillator and measured applied external voltage (V_{ext}) to time in case of $V_{ext} = 1.46$ V. b V_{ext} vs time plot magnifying the region from 5 us to 30 us (gray-colored region in a). c Histogram of V_{ext} . d Power spectral density (PSD) plot of V_{ext} to frequency, showing the relationship between PSD and frequency, indicating that the closer a frequency is to zero, the closer it is to white noise.

Supplementary Fig. S4a depicts the simplified NbO_x oscillator circuit diagram, along with a measured applied external voltage (V_{ext}) over time at $V_{ext} = 1.46$ V. Supplementary Fig. S4b zooms into the region between 5 us and 30 us, further elucidating the noise behavior. Additionally, Supplementary Fig. S4c presents a histogram of the V_{ext} data from Supplementary Fig. S4b, revealing a normal distribution with a median of 1.443 V and a standard deviation (stdev) of 0.0023 V. In Supplementary Fig. S4d, we conduct a noise element analysis by plotting the power spectral density (PSD) to frequency. The linear-fitted function shows that the PSD have a function of the power of 0.0323 of the frequency, which is close to 0, confirming that the injected voltage noise is the white noise. This analysis aligns with our overall findings and supports our introduction of white noise in p-osc model.

In addition, we adressed it on page 6 in main text as follows.

Page 6 in main text

“... We introduced a time-dependent OU process since white noises were involved in the thermal and electrical conditions of our device (Supplementary Fig. S4). ...”

In response to the reviewer's comment about the relatively high noise amplitude for white noise, It's worth noting that the noise level we observed in our measurement system was around 2.3 mV at external voltage of 1.46 V. This observed value could potentially be higher than the actual noise amplitude affecting the device due to various factors such as noise figure or noise factor of the amplifier, impact of gain, non-linearity of circuit, and interaction with external noise²². Given these considerations, we have conservatively set the electrical noise at 0.1 mV, which is significantly lower than what we measured, to account for these potential inaccuracies in our measurement system.

[R3_#6] 6. The manuscript shows a probabilistic computing demonstration using a NbOx-based Boltzmann machine. However, I don't find any description of the actual implementation of the Boltzmann machine in the manuscript. Such a description is necessary (how many actual NbOx devices are used, how signals are collected and processed, etc.). I am especially interested in how the different NbOx oscillators are synchronized with each other, as the operation scheme described in the manuscript requires such synchronization. Furthermore, the estimation of energy consumption for the MVC task is desirable.

>> **[Author Response]**

Thank you for the valuable comment. In order to more accurately convey the operation of the MBM in our study, we have supplemented the description of MBM on page 8 in main text as follows.

Page 8 in main text

“Here, we implemented a Boltzmann machine³² adopting the established p-osc model-based pbits (NbO_x pbit) for solving NP-hard problems through simulations. ...”

In addition, our implementation follows a sequential updating process for each pbit, rendering explicit synchronization unnecessary, meaning that each oscillators are totally electrically seperated. We added more detailed description on the MBM to solving the MVC problem in the Supplementary Information section 9 as follows.

Supplementary Information section 9

9. In-depth description of the MBM to solving MVC problem

As described in the manuscript, in the operation of the Boltzmann machine, the negative derivative of the Hamiltonian ($h(\vec{X})$) for each pbit at the current time step ($D_i^t = -\partial h(\vec{X}^t)/\partial X_i$) is used as the input value for that pbit at the next time step. However, as can be seen from the $h(\vec{X})$ in Equation 10 of main text, D_i^t depends on the absolute size of the $h(\vec{X})$ which is affected by α and β (or additional parameters in other problems⁵) or the way the developed $h(\vec{X})$ is defined. Therefore, when simply using D_i^t as an input value for the next step, an excessively large or small D_i^t may make the stochastic behavior too deterministic, or may not induce significant differences in others. Thus, to mitigate this problem previously, we arithmetically calculated the maximum and minimum values of D_i^t that could

be obtained from the initially set $h(\vec{X})$ and normalized D_i^t into the device operating range V_{ext} .

In case of minimum vertex covering (MVC) problem⁵, $h(\vec{X})$ as quadratic unconstrained binary optimization (QUBO) formulation is described as Equation S20 (same as Equation 10 of main text), where $X_v \in \{0, 1\}$.

$$h(\vec{X}) = \alpha \sum_{u,v \in E} (1 - X_u)(1 - X_v) + \beta \sum_{v \in V} X_v \quad (\text{S20})$$

If we rewrite this equation in vector-matrix form, $h(\vec{X})$ is described as Equation S21.

$$h(\vec{X}) = \alpha \vec{X}^T (0.5A - D) \vec{X} + \alpha \mathbb{1}^T (0.5A) \mathbb{1} + \beta \vec{X}^T \vec{X} \quad (\text{S21})$$

$$h(\vec{X}) = X^T \{ \alpha(0.5A - D) + \beta I \} X + 2\alpha \quad (\text{S22})$$

where A is adjacency matrix, D is degree matrix, $\mathbb{1}$ is all-ones matrix of the same size as A and D , and I is identity matrix. If we set each element in $(0.5A - D)$ as P_{ij} , Equation S22 is describes as below:

$$h(\vec{X}) = \vec{X}^T \begin{pmatrix} \alpha P_{11} + \beta & \alpha P_{12} & \cdots & \alpha P_{1n} \\ \alpha P_{21} & \alpha P_{22} + \beta & \cdots & \alpha P_{2n} \\ \vdots & \vdots & \ddots & \vdots \\ \alpha P_{n1} & \alpha P_{n2} & \cdots & \alpha P_{nn} + \beta \end{pmatrix} \vec{X} + 2\alpha \quad (\text{S23})$$

Due the symmetric property of adjacency matrix A ,

$$h(\vec{X}) = \vec{X}^T \begin{pmatrix} \alpha P_{11} + \beta & \alpha P_{12} & \cdots & \alpha P_{1n} \\ \alpha P_{12} & \alpha P_{22} + \beta & \cdots & \alpha P_{2n} \\ \vdots & \vdots & \ddots & \vdots \\ \alpha P_{1n} & \alpha P_{2n} & \cdots & \alpha P_{nn} + \beta \end{pmatrix} \vec{X} + 2\alpha = \vec{X}^T Q \vec{X} + 2\alpha \quad (\text{S24})$$

While calculation of $h(\vec{X})$, $X_v^2 = X_v$ due to $X_v \in \{0, 1\}$. Thus, $\partial h(\vec{X}^t) / \partial X_i$ can be simply described as below equation, which aligns with neuronal dynamics between neurons⁶.

$$\frac{\partial h(\vec{X})}{\partial X_i} = Q_{ii} + \sum_{j=1, j \neq i}^n 2Q_{ij} X_j \quad (\text{S25})$$

The off-diagonal elements of Q is equal to corresponding element of $0.5A$ and the adjacency matrix A has only positive value. Thus, the input value of the i^{th} pbbit for the next time step can have a range of values as below:

$$-\frac{\partial h(\vec{X})}{\partial X_i} \in \left[-Q_{ii}, -\left(Q_{ii} + \sum_{j=1, j \neq i}^n 2Q_{ij} \right) \right] \quad (\text{S26})$$

So, from the final Equation S26, we can get n sets of maximum and minimum values of $-\frac{\partial h(\vec{X})}{\partial X_i}$ for n pbbit systems. We then take the absolute values of all n sets and designated the largest absolute value among the $2n$ values as D_{max} and $D_{\text{min}} = -D_{\text{max}}$.

In case of graph G(6, 7) in Fig. 3b, the A , D , and Q matrix is defined as below.

$$A = \begin{pmatrix} 0 & 1 & 0 & 0 & 0 & 1 \\ 1 & 0 & 1 & 0 & 0 & 1 \\ 0 & 1 & 0 & 1 & 1 & 0 \\ 0 & 0 & 1 & 0 & 0 & 0 \\ 0 & 0 & 1 & 0 & 0 & 1 \\ 1 & 1 & 0 & 0 & 1 & 0 \end{pmatrix}, D = \begin{pmatrix} 2 & 0 & 0 & 0 & 0 & 0 \\ 0 & 3 & 0 & 0 & 0 & 0 \\ 0 & 0 & 3 & 0 & 0 & 0 \\ 0 & 0 & 0 & 1 & 0 & 0 \\ 0 & 0 & 0 & 0 & 2 & 0 \\ 0 & 0 & 0 & 0 & 0 & 3 \end{pmatrix}, \quad (S27)$$

$$Q = \begin{pmatrix} \beta - 2\alpha & 0.5\alpha & 0 & 0 & 0 & 0.5\alpha \\ 0.5\alpha & \beta - 3\alpha & 0.5\alpha & 0 & 0 & 0.5\alpha \\ 0 & 0.5\alpha & \beta - 3\alpha & 0.5\alpha & 0.5\alpha & 0 \\ 0 & 0 & 0.5\alpha & \beta - \alpha & 0 & 0 \\ 0 & 0 & 0.5\alpha & 0 & \beta - 2\alpha & 0.5\alpha \\ 0.5\alpha & 0.5\alpha & 0 & 0 & 0.5\alpha & \beta - 3\alpha \end{pmatrix}$$

Then, the minimum, maximum value set for each pbit can be derived as below equations. We have set $\alpha = 2$ and $\beta = 1$ in this research.

$$-\frac{\partial h(\vec{X})}{\partial X_1} \in [-1,3] \quad (S28)$$

$$-\frac{\partial h(\vec{X})}{\partial X_2} \in [-1,5] \quad (S29)$$

$$-\frac{\partial h(\vec{X})}{\partial X_3} \in [-1,5] \quad (S30)$$

$$-\frac{\partial h(\vec{X})}{\partial X_4} \in [-1,1] \quad (S31)$$

$$-\frac{\partial h(\vec{X})}{\partial X_5} \in [-1,3] \quad (S32)$$

$$-\frac{\partial h(\vec{X})}{\partial X_6} \in [-1,5] \quad (S33)$$

Thus, D_{\max} and D_{\min} are designated as 5 and -5 in this case."

Futhermore, we have updated the energy calculation parts of ***pages 12 in main text*** as follows. In this part, the (number of iterations to solution) is the number of iterations actually performed during the MVC solving process with MBM. Therefore, the energy calculation in this part is for the MVC task and we have clarified this.

Pages 12 in main text

"Lastly, we have estimated the energy consumption of pbits for the MBM operation in solving the MVC problem. Here, the total energy consumption of pbits (energy to solution) can be approximately defined by (number of pbits, n) \times (energy during bit generation, E_{pbit}) \times (number of iterations to solution, N). ..."

[R3_#7] 7. Authors should clarify how the device performance of “260 ns with an energy of 114 pJ bit⁻¹” is derived because the energy consumption is expected to vary depending on the computational task.

>> **[Author Response]**

Thank you for the comment about clarification of energy consumption of NbO_x pbit. 260 ns is the oscillation period, and 114 pJ is the energy of one oscillation. During non-oscillating period, it consumes energy of 141 pJ. Each oscillation or non oscillation during period is bit generation and these are the basic characteristics of the NbO_x pbits which is not depends on different computational task.

However, we agree that the previous description of the energy consumption during oscillation may give a mislead as it is a representative value of the energy consumption during stochastic behavior (which depends on the computational task), so in the revised manuscript we have specified the energy consumption during oscillation and non-oscillation situations.

Page 4 in main text

“... The probabilistic oscillation generates probabilistic bits in 260 ns with an energy of 114 pJ per oscillation or 141 pJ per non-oscillation (staying in the metallic state), ...”

Also, as mentioned by the reviewer, the energy consumption in pbit during MBM operation depends on the computational task. In this case, the total energy consumption of pbits can be calculated by (number of pbits, n) \times (energy during bit generation, E_{pbit}) \times (number of iterations to solution, N) as our response in [R3_#6]. However, during MBM operation process, the oscillation probability of each pbit is constantly changing, and given its stochastic nature, it undergoes a different behavior process for each trial. Thus, we use the average energy consumption by assuming $p_{\text{osc}} = 0.5$ and described it in page 12 in main text. However, we would appreciate if you could note that this is not an exact value for the task due to the stochastic evolution nature of the Boltzmann machine, and that it is meaningful to compare for each Δ case.

Page 12 in main text

“... Here, E_{pbit} is the energy required for generating the pbit’s output. To obtain the exact E_{pbit} , it is necessary to count the occurrence of oscillation and non-oscillation cases, which is very complicated. So, we assumed that each case occurs with a 50% probability. Then, E_{pbit} can be set to $128 \times (\Delta + 1)$ pJ/bit, where 128 pJ is the average energy per oscillation or non-oscillation and $(\Delta + 1)$ is a factor by the delay period. ...”

[R3_#8] 8. In Figs. 2a and 2b, the simulation is done with the load line inside the NDR-2 hysteresis, and probabilistic oscillation behavior is observed. However, the measurement results in Figs. 1a (red curve) and 1d show the probabilistic oscillations even though the load line is outside the NDR-2 hysteresis (where I expect continuous oscillations rather than probabilistic oscillations). What is the reason for this unexpected observation?

>> **[Author Response]**

Thank you for providing the opportunity to clarify the conditions under which probabilistic oscillations occur in our NbO_x oscillator system. To address your comment, we have developed a new simulation model that closely reflects the real-world measurement system used in our experiments. We added the result to the Supplementary Information section 3 and referred it in the page 5 in main text as follows

Supplementary Information section 3

“3. Detailed description of the measurement system’s effect on device characterization

Fig. S3. NbO_x oscillator measurement system. **a** A schematic of measurement system with Keithley 4200 SCS semiconductor analyzer and dark box. **b** Equivalent circuit diagram of **a**

The measurement system we used to evaluate the NbO_x oscillator is illustrated in Fig. S3a. Two PMU cards connected to a Keithley 4200 SCS semiconductor analyzer apply voltage to both terminals of the NbO_x memristor and read the output. PMU 1 is connected to a series resistor module, and the probe makes contact with the top electrode of the NbO_x memristor. Another probe contacts the bottom electrode and is directly connected to PMU 2. All components, except for the semiconductor analyzer, are located within a dark box.

Based on this setup, an equivalent NbO_x oscillator circuit is presented in Fig. S3b. Circuit elements corresponding to components in the measurement system are colored to match those in Fig. S3a.

	R_L	$R_{TE} + R_{cont}$	$R_{BE} + R_{cont}$	R_{osc}	C_p	R_{total}	V_{ext}	Oscillate?
Unit	Ω	Ω	Ω	Ω	pF	Ω	V	-
Case 1 (Reference)	1200	0	0	0	0	1200	1.32	Yes
							1.33	Yes
Case 2	1200	0	0	0	20	1200	1.32	Yes
							1.33	No
Case 3	1100	50	50	0	20	1200	1.32	No
							1.33	No

Table S1. Simulation parameters of NbO_x oscillator measurement system

We used a noise-free NbO_x memristor model for the simulation, and the results under various conditions are summarized in Table S1.

For case 1, we configured the oscillator circuit identically to the one described in the main text, setting only the load resistor (R_L) to 1200 Ω and removing all other elements. When a DC voltage (V_{ext}) of 1.32 V and 1.33 V was applied, we observed oscillations in the output current, consistent with Fig. 2d in the main text.

In case 2, we introduced a parasitic capacitance (C_p) to reflect the electrically isolated state of the probe in the measurement system. As a result, the node voltage at the front end of the NbO_x device becomes time-dependent due to C_p . Under these conditions, we observed oscillations at $V_{ext} = 1.32$ V but not at 1.33 V, despite the total series resistance being the same 1200 Ω as in case 1. This indicates that C_p influences the oscillation conditions of the NbO_x oscillator. In particular, it can be seen that the non-oscillation of $V_{ext} = 1.33$ V, despite the circuit being configured with the same series resistor, is entirely an effect of C_p compared to case 1.

This trend is even more severe in case 3, which incorporates the resistances of the top and bottom electrodes as well as the contact resistance of the probe, thus closely mimicking the actual experimental setup. In this case, the circuit did not oscillate even at $V_{ext} = 1.32$ V, so it can be seen that the effect of voltage dividing due to the series resistances further increases the influence of the C_p on oscillation.

From these comparative simulation results, in a real-world experimental setup, the presence of C_p has the effect of making the system to converge rather than oscillate from a lower point than when calculated considering only the equilibrium (static) state ($i_L = i_{osc} = \frac{V_{ext} - V_m}{R_L}$) of the system. Consequently, even though the equilibrium point is formed at a point slightly below the NDR-2 region, as shown in the red curves in Figure 1a, it can be understood that probabilistic oscillations can occur in the measurement environment due to the resulting non-oscillatory state caused by C_p .”

Page 5 in main text

“... More details on the measurement system configuration and its influence on the device characterization can be found in Supplementary Information section 3.”

Reference

1. Hertz, J. *et al.* The Hopfield Model. *Introd. to Theory Neural Comput.* 11–41 (2018).
2. Ackley, D. H., Hinton, G. E. & Sejnowski, T. J. A learning algorithm for boltzmann machines. *Cogn. Sci.* **9**, 147–169 (1985).
3. Borders, W. A. *et al.* Integer factorization using stochastic magnetic tunnel junctions. *Nature* **573**, 390–393 (2019).
4. Camsari, K. Y., Salahuddin, S. & Datta, S. Implementing p-bits with Embedded MTJ. *IEEE Electron Device Lett.* **38**, 1767–1770 (2017).
5. Sutton, B., Camsari, K. Y., Behin-Aein, B. & Datta, S. Intrinsic optimization using stochastic nanomagnets. *Sci. Rep.* **7**, 44370 (2017).
6. Faria, R., Camsari, K. Y. & Datta, S. Low-Barrier Nanomagnets as p-Bits for Spin Logic. *IEEE Magn. Lett.* **8**, 4105305 (2017).
7. Zhou, Y. & Ramanathan, S. Mott Memory and Neuromorphic Devices. *Proc. IEEE* **103**, 1289–1310 (2015).
8. Zhang, X. *et al.* An artificial spiking afferent nerve based on Mott memristors for neurorobotics. *Nat. Commun.* **11**, 51 (2020).
9. Kumar, S., Strachan, J. P. & Williams, R. S. Chaotic dynamics in nanoscale NbO₂ Mott memristors for analogue computing. *Nature* **548**, 318–321 (2017).
10. Pickett, M. D., Medeiros-Ribeiro, G. & Williams, R. S. A scalable neuristor built with Mott memristors. *Nat. Mater.* **12**, 114–117 (2013).
11. Kumar, S., Williams, R. S. & Wang, Z. Third-order nanocircuit elements for neuromorphic engineering. *Nature* **585**, 518–523 (2020).
12. Zhang, J. *et al.* Thermally induced crystallization in NbO₂ thin films. *Sci. Rep.* **6**, 34294 (2016).
13. Bolzan, A. A., Fong, C., Kennedy, B. J. & Howard, C. J. Structural Studies of Rutile-Type Metal Dioxides. *Acta Crystallogr. Sect. B Struct. Sci.* **53**, 373–380 (1997).
14. Naito, K., Kamegashira, N. & Sasaki, N. Phase equilibria in the system between NbO₂ and Nb₂O₅ at high temperatures. *J. Solid State Chem.* **35**, 305–311 (1980).
15. Chudnovskii, F. A., Odynets, L. L., Pergament, A. L. & Stefanovich, G. B. Electroforming and switching in oxides of transition metals: The role of metal-insulator transition in the switching mechanism. *J. Solid State Chem.* **122**, 95–99 (1996).
16. Kim, G. *et al.* Self-clocking fast and variation tolerant true random number generator based on a stochastic mott memristor. *Nat. Commun.* **12**, 2906 (2021).
17. Park, W. *et al.* High Amplitude Spike Generator in Au Nanodot-Incorporated NbO_x Mott Memristor. *Nano Lett.* **23**, 5399–5407 (2023).
18. Messaris, I. *et al.* A simplified model for a NbO₂ Mott memristor physical realization. in *2020 IEEE International Symposium on Circuits and Systems (ISCAS)* 1–5 (2020).
19. Kumar, S. *et al.* Physical origins of current and temperature controlled negative differential resistances in NbO₂. *Nat. Commun.* **8**, 658 (2017).
20. Pickett, M. D. & Stanley Williams, R. Sub-100fJ and sub-nanosecond thermally driven threshold switching in niobium oxide crosspoint nanodevices. *Nanotechnology* **23**, 215202

(2012).

21. Kay, A. *Operational Amplifier Noise*. (Newnes, 2012).
22. Leach, W. M. Fundamentals of Low-Noise Analog Circuit Design. *Proc. IEEE* **82**, 1515–1538 (1994).

REVIEWERS' COMMENTS

Reviewer #1 (Remarks to the Author):

Authors addressed the questions and revised the manuscript accordingly. Now this is in a good shape to be published.

Reviewer #2 (Remarks to the Author):

The authors addressed my comments.

Reviewer #3 (Remarks to the Author):

The authors took the reviewer's comments seriously and revised the manuscript accordingly. I believe the revised manuscript is significantly improved in clarity and significance.

1. If possible, showing the NbO_x pbit characteristics (such as *I-V* curves) as a function of the junction area would further help regarding the core-shell picture in the authors' reply to the 4th point of this reviewer's previous report.

2. Furthermore, some discussions are desirable regarding the energy consumption necessary to calculate the pbit inputs $V_{\text{ext},i}$ from the previous outputs in the digital domain, as this will be a significant energy burden when scaling to many pbits. (Related to the 7th point of this reviewer's comment.)

3. In the introduction, the authors compare the NbO_x pbits to the conventional pbits. Please add some description or reference because it is unclear what the conventional pbits mean.

Other than these points, I recommend publication in Nature Communications.

REVIEWERS' COMMENTS

Reviewer #1 (Remarks to the Author):

Authors addressed the questions and revised the manuscript accordingly. Now this is in a good shape to be published.

Reviewer #2 (Remarks to the Author):

The authors addressed my comments.

Reviewer #3 (Remarks to the Author):

The authors took the reviewer's comments seriously and revised the manuscript accordingly. I believe the revised manuscript is significantly improved in clarity and significance.

>> [Author Response]

Thank you for your thoughtful comments. In the following, we have carefully reviewed the additional comments and prepared one-by-one responses. We hope that these adequately address your concerns.

[R3_#1] 1. If possible, showing the NbO_x pbit characteristics (such as I-V curves) as a function of the junction area would further help regarding the core-shell picture in the authors' reply to the 4th point of this reviewer's previous report.

>> [Author Response]

We would like to extend our sincere gratitude for your valuable comment on internal structure of NbO_x device. In response to the reviewer's request, we have included the data in Supplementary Figs. S1c and S1d. Furthermore, we have made the following revisions in the revised manuscript.

Supplementary Information section 1

"1. Area-dependent device capacitance and current characteristics

Fig. S1. **a** C-V measurement result of NbO_x memristor with a diameter of 40 nm. **b** Capacitances of NbO_x memristors with different diameters (100 nm, 60 nm, 40 nm) before (pristine) and after electroforming. **c** NDR behaviors with different diameters (100 nm, 60 nm, 40 nm) after electroforming. They exhibited almost identical NDR behaviors considering the area variation, especially at the off state. **d** Device conductances before (pristine) and after electroforming at the off state read (at -0.1 V). These results strongly support the core-shell model, where the localized core dominates the NDR behaviors.

Some variations in NDR behaviors were noted, likely attributable to shell characteristics, including shell area and conductivity.

”

Page 4 in main text

“... The constant device capacitance and almost identical I - V characteristics after electroforming with respect to the device’s area (Supplementary Fig. S1) suggested that the switching is associated with a localized region, following a core-shell model^{15–18,20,23,24}. ...”

[R3_#2] 2. Furthermore, some discussions are desirable regarding the energy consumption necessary to calculate the pbit inputs $V_{text,i}$ from the previous outputs in the digital domain, as this will be a significant energy burden when scaling to many pbits. (Related to the 7th point of this reviewer's comment.)

>> [Author Response]

Thank you for your exact comment. As the reviewer indicated, we omitted the energy consumption portion in memory and processor in our energy calculation process. We have made the following revision to provide the rationale for this omission.

Page 12 in main text

“... In the energy consumption calculation, we considered only the energy consumption in p-bits and excluded the energy consumed in memory (storing output vectors) and process units (calculating the Hamiltonian gradient and obtaining the subsequent input voltages from it), as these parts are commonly required in p-computing and are handled by conventional digital computers. ...”

[R3_#3] 3. In the introduction, the authors compare the NbOx pbits to the conventional pbits. Please add some description or reference because it is unclear what the conventional pbits mean.

>> [Author Response]

Thank you for pointing out the deficiencies in our text. Following this comment, we added the following references accordingly.

Page 3 in main text

“Moreover, differing from conventional pbits^{9,21,22}, ...”

Other than these points, I recommend publication in Nature Communications.

Reference

1. Woo, K. S. *et al.* Probabilistic computing using Cu_{0.1}Te_{0.9}/HfO₂/Pt diffusive memristors. *Nat. Commun.* **13**, 5762 (2022).
2. Choi, S. *et al.* Controllable SiO_x Nanorod Memristive Neuron for Probabilistic Bayesian Inference. *Adv. Mater.* **34**, 2104598 (2022).

3. Gaba, S., Sheridan, P., Zhou, J., Choi, S. & Lu, W. Stochastic memristive devices for computing and neuromorphic applications. *Nanoscale* **5**, 5872 (2013).